# REvolve: Reward Evolution with Large Language Models using Human Feedback

**Rishi Hazra,**[*] **Alkis Sygkounas,**[*] **Andreas Persson,**
**Amy Loutfi & Pedro Zuidberg Dos Martires**
Centre for Applied Autonomous Sensor Systems (AASS)
Örebro University, Sweden
{rishi.hazra, alkis.sygkounas, andreas.persson, amy.loutfi,
pedro.zuidberg-dos-martires}@oru.se

https://rishihazra.github.io/REvolve

## Abstract

Designing effective reward functions is crucial to training reinforcement learning (RL) algorithms. However, this design is non-trivial, even for domain experts, due to the subjective nature of certain tasks that are hard to quantify explicitly. In recent works, large language models (LLMs) have been used for reward generation from natural language task descriptions, leveraging their extensive instruction tuning and commonsense understanding of human behavior. In this work, we hypothesize that LLMs, guided by human feedback, can be used to formulate reward functions that reflect human implicit knowledge. We study this in three challenging settings – autonomous driving, humanoid locomotion, and dexterous manipulation – wherein notions of "good" behavior are tacit and hard to quantify. To this end, we introduce REvolve, a truly evolutionary framework that uses LLMs for reward design in RL. REvolve generates and refines reward functions by utilizing human feedback to guide the evolution process, effectively translating implicit human knowledge into explicit reward functions for training (deep) RL agents. Experimentally, we demonstrate that agents trained on REvolve-designed rewards outperform other state-of-the-art baselines.

## 1 Introduction

Recent successes in applying reinforcement learning (RL) have been concentrated around areas with clearly defined reward functions (Silver et al., 2018; Vinyals et al., 2019; Fawzi et al., 2022; Mankowitz et al., 2023; DeepMind, 2024). However, tasks with ambiguously defined rewards, such as autonomous driving, still present challenges due to their complex and difficult-to-specify goals (Knox et al., 2023). This echoes Polanyi's paradox, which states that *we know more than we can tell* (Polanyi, 2009). The notions of "good" behavior (i.e., aligned with human standards) are inherently subjective – involving a level of tacit knowledge that is hard to quantify and encode into reward functions. This is also referred to as the *reward design problem* and often leads to sub-optimal behaviors that exploit shortcuts in the reward function and that do not truly reflect human preferences (Kerr, 1975; Hadfield-Menell et al., 2017; Booth et al., 2023). This issue is underscored by the growing concerns about the misalignment between human values and the objectives pursued by agents (Turner et al., 2021; Omohundro, 2018; Russell, 2019).

Recently, LLMs have shown impressive abilities in instruction following (Longpre et al., 2023; Peng et al., 2023), search and black-box optimization (Yang et al., 2024; Zhang et al., 2023; Liu et al., 2024), self-reflection (Madaan et al., 2023; Shinn et al., 2023), and code generation (Chen et al., 2021; Roziere et al., 2023). Owing to their vast training and world knowledge (Wang et al., 2023; Hao et al., 2023; Hazra et al., 2024), they can readily incorporate commonsense priors of human behavior. Together, these abilities make LLMs an ideal candidate for generating rewards from natural language (NL) task descriptions and refining them using language-based feedback loops. Indeed,

---

[*]Equal contribution.

recent works like Language to Rewards (Yu et al., 2023) and Text2Reward (Xie et al., 2024) have utilized LLMs for this purpose. More significantly, Eureka (Ma et al., 2024a) utilizes LLMs to design dense and interpretable reward functions, achieving improved performance in dexterous manipulation tasks. It does so by generating multiple reward functions and *greedily* mutating the best candidate over multiple iterations. As such, Eureka resembles more of a *greedy iterative* search than an evolutionary search, as described in Ma et al. (2024a).

We identify this greedy approach as a significant limitation that risks premature convergence and leads to a waste of resources (by preemptively discarding *weaker* reward functions). Moreover, Eureka relies on having access to fitness functions to assess the quality of learned behaviors and decide the best reward function. This leads to a chicken-and-egg dilemma – if designing a good fitness measure was feasible, one could arguably design an effective reward function just as easily. This is particularly challenging for complex tasks, such as autonomous driving or dexterous manipulation.

To this end, we introduce **R**eward **E**volve (REvolve), a novel evolutionary framework using LLMs, specifically GPT-4, to output reward functions (as executable Python codes) and evolve them based on human feedback. This allows humans to serve as fitness functions by mapping their preferences and feedback into fitness scores.

In particular, REvolve offers the following novelties:

**(1) Using Evolutionary Algorithms (EAs) for Reward Design.** Traditional gradient-based optimization is unsuitable for reward design due to the lack of a differentiable cost function. Instead, REvolve utilizes meta-heuristic optimization through EAs (De Jong, 2017) to search for the best candidates. This is done by evolving a population of reward functions using genetic operators like mutation, crossover, and selection – employing LLMs as intelligent operators instead of traditional handcrafted methods. We empirically show that our evolutionary framework considerably outperforms iterative frameworks like Eureka – all *without incurring additional computational costs compared to Eureka*. Through REvolve, we also demonstrate that EAs, combined with the full range of operations like mutation, crossover, selection, and migration, can be successfully implemented with LLMs.

**(2) Utilizing Human Feedback to Guide the Search.** Mapping human notions directly into explicit reward functions is challenging; however, humans can still evaluate learned behaviors, effectively acting as fitness functions. Through REvolve, human preferences are directly mapped into fitness scores. Additionally, REvolve incorporates qualitative natural language feedback, offering GPT-4 high-level insights into which aspects to prioritize while refining the reward functions. Together, this ensures that the reward functions – and thus the learned policies – more closely reflect human notions.

**(3) Eliminating the Need for Additional Reward Model Training.** Unlike reinforcement learning with human feedback (RLHF) (Christiano et al., 2017), which relies on a black-box reward model – trained on massive amounts of curated human preference data – to align LLMs with goals like safety and reliability (OpenAI, 2022), REvolve requires no extra models. *REvolve-designed reward functions are interpretable* (as executable Python code) compared to RLHF rewards, making it more suitable for safety-critical applications, such as autonomous driving (Christian, 2023). We also include an estimated data requirement comparison in Appendix D.

Figure 1, provides an overview of REvolve for autonomous driving within a simulated environment.

## 2 Problem Formalization

Let us first formally introduce the reward design problem (RDP) (Singh et al., 2009), given by the tuple $\langle M, \mathcal{R}, \pi, F \rangle$, where $M = (S, A, T)$ is the world model consisting of state space $S$, action space $A$, and transition function $T$. $\mathcal{R}$ is the set of all reward functions where each $R \in \mathcal{R}$ is a reward function that maps a state-action pair to a real-valued scalar $R : S \times A \to \mathbb{R}$. The tuple $(M, R)$ constitutes a Markov decision process (Sutton & Barto, 2018). Furthermore, $\pi : S \to \Delta A$ is a policy that maps a state to a distribution over actions, which is learned using a reward function $R$.

Moreover, the RDP includes a function $F$ that evaluates the fitness (i.e., quality) of the trained policies. Traditionally, $F$ maps policy rollouts, i.e., rollouts of agent-environment interactions, to real-valued scalars. Formally, we denote this by $F : \Omega(\Theta_\pi) \to \mathbb{R}$, where $\Theta_\pi$ is the random variable modeling the probability distribution of rollouts induced by $\pi$ and $\Omega(\Theta_\pi)$ is its sample space.

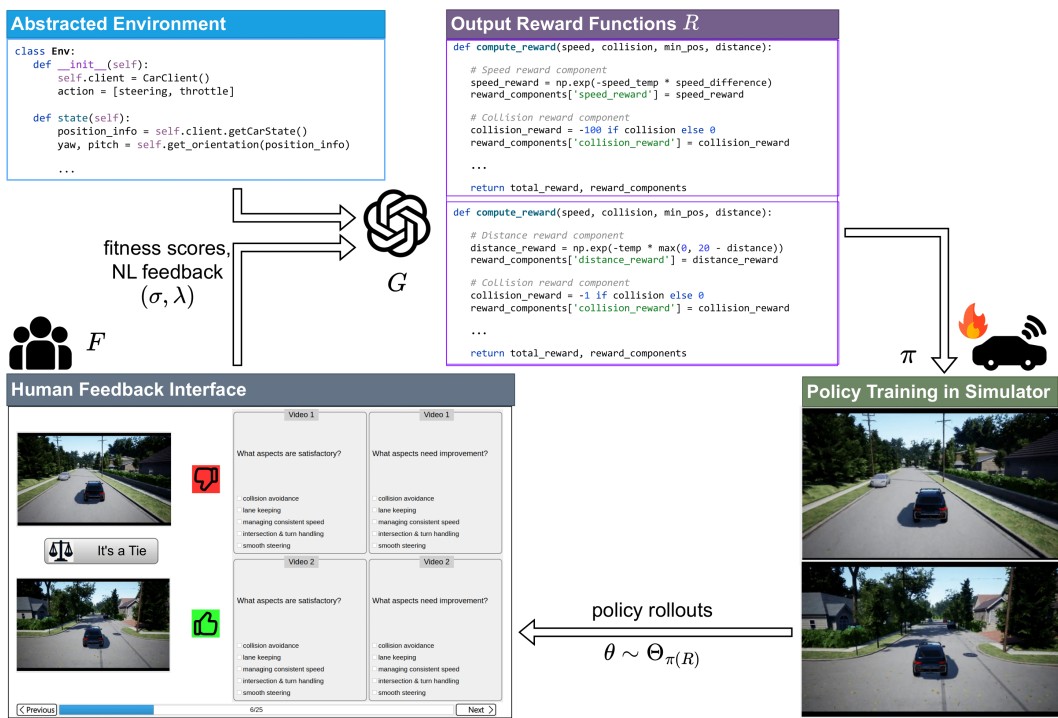

Figure 1: An overview of REvolve. Given the task (here, autonomous driving) and abstracted environment variables, a reward designer $G$ (LLM) outputs a population of reward functions $R \in \mathcal{R}$, each used to train a policy $\pi(R)$ in simulation. Then, we collect human preferences and natural language (NL) feedback on pairs of policy rollouts $\theta \sim \Theta_{\pi(R)}$ through a user feedback interface. Policy (and thus, corresponding reward function) fitness $\sigma$ is calculated, and the fittest individuals, along with their NL feedback $\lambda$, are refined by $G$. In addition, the process leverages genetic programming for evolution.

For REvolve, we generalize this to also include natural language feedback. Specifically, we redefine $F : \Omega(\Theta_\pi) \to \mathbb{R} \times \mathcal{L}$, where $\mathcal{L}$ is the set of all strings (including natural language feedback). Note that the fitness function is derived from human evaluations of the trained policies, thereby rendering it non-differentiable.

In order to instantiate the RDP tuple for REvolve, we assume that we are given a reward designer $G : \mathcal{L} \to \mathcal{R}$ that maps from the set of all strings to the set of all reward functions $\mathcal{R}$. The objective now is to find a reward function $R \in \mathcal{R}$ that maximizes the expected fitness score of the policy $\pi$ trained on $R$, which we denote by $\pi(R)$. Formally, we express the optimization problem as follows:

$$R^* \coloneqq \underset{R \in \mathcal{R}}{\arg\max}\Big[\mathbb{E}_{\theta \sim \Theta_{\pi(R)}}\left[F(\theta)\right]\Big] \tag{1}$$

Here, $\theta$ is a trajectory sampled from $\Theta_{\pi(R)}$. We also define the optimal policy $\pi^*$ as the policy trained with reward function $R^*$ given as $\pi^* \coloneqq \pi(R^*)$.

An implicit assumption for reward design is the knowledge of observation and action variables in the environment. For real-world problems, like autonomous driving, the assumption is supported by access to sensor data (such as speed from speedometers and distance covered from odometers) or by employing advanced vision-based models (Kirillov et al., 2023; Cheng et al., 2024; Yuan et al., 2022).

## 3 REWARD EVOLUTION

As the fitness function $F$ is based on implicit human understanding, it cannot be explicitly formulated and is, therefore, not amenable to gradient-based optimization. Hence, we solve the RDP using a meta-heuristic approach through genetic programming (GP) (Langdon & Poli, 2013). Concretely,

---

**Algorithm 1** REvolve

---

1: **Require** fitness function $F$, reward designer $G$
2: **Hyperparameters** evolution generations $N$, number of sub-populations $I$, individuals per generation $K$, mutation probability $p_m$, crossover probability $p_c$
3: **Input:** Task description
4: # Initialization                                                                               ▷ Section 3.1
5: $D := \{(R, \pi, \sigma, \lambda)_i\}_{i=1}^K$                                                  ▷ Initialize database
6: **while** not termination **do**
7:     # Reproduction                                                                             ▷ Section 3.2
8:     $D_{\text{temp}} := \{\}$                                                                   ▷ Initialize empty temporary database
9:     **for** $K$ individuals **do**
10:         mutation $\sim$ Bernoulli($p_m$)                                                       ▷ Sample operation
11:         $P \sim$ WeightedSample($D, \{\sigma_j^P\}_{j=1}^I$) ▷ Sample sub-population using average fitness
12:         **if** mutation **then**
13:             $R_{\text{new}} :=$ Mutate($G, D[P]$)                                              ▷ Mutate individual in sub-population
14:         **else**
15:             $R_{\text{new}} :=$ Crossover($G, D[P]$)                                           ▷ Crossover with $1 - p_m$ probability
16:         **end if**
17:         $\pi :=$ Train($R_{\text{new}}$)                                                       ▷ Train policy on new reward
18:         $D_{\text{temp}} \leftarrow$ Update($R_{\text{new}}, P, \pi$)                          ▷ Add to temporary database
19:     **end for**
20:     # Selection                                                                               ▷ Section 3.3
21:     **for** $(R, P, \pi) \in D_{\text{temp}}$ **do**
22:         $\theta \sim \Theta_\pi$                                                               ▷ Sample rollouts from trained policy
23:         $(\sigma, \lambda) := F(\theta)$                                                       ▷ Compute fitness from human feedback
24:         **if** $\sigma \geq \sigma^P$ **then**                                ▷ If individual fitness $\geq$ sub-population fitness
25:             $D[P] \leftarrow$ Update($R, \pi, \sigma, \lambda$)                                ▷ Add reward to database sub-population
26:         **end if**
27:     **end for**
28: **end while**
29: # Termination                                                                                 ▷ Section 3.4
30: $(R^*, \pi^*, \sigma^*, \lambda^*) := \arg\max_\sigma D$          ▷ Select tuple with the highest fitness score $\sigma$
31: **Return** $\pi^*$

---

REvolve uses GP to iteratively refine a population of reward functions (individuals) by applying genetic operations such as mutation, crossover, and migration. In each iteration, new individuals are obtained and added to the population based on their fitness. This fitness is assessed through human evaluations of the policies trained with these reward functions – thus guiding the evolutionary search.

Evolutionary algorithms, in general, are considered to be weak methods that perform blind searches and do not exploit domain-specific knowledge (De Jong, 2017). The initialization and perturbations are random, leading to scalability issues (O'Neill et al., 2010). In contrast, LLMs can leverage their commonsense priors to suggest changes that are intelligent and better aligned with the feedback (Gugerty & Olson, 1986). Therefore, we combine the evolutionary search of REvolve with LLMs by using them as reward designers $G$ in the RDP.

We follow the island model of evolution (Cantú-Paz et al., 1998), where we maintain multiple ($I$) sub-populations (or islands) in a reward database $D$. Island populations evolve by locally reproducing (mutation and crossover). Periodically, individuals migrate from one island to another. The island model helps enhance genetic diversity across the population and prevent premature convergence in complex search spaces (Cantú-Paz et al., 1998). Following existing works in reward design with LLMs (Xie et al., 2024; Ma et al., 2024a), we represent the reward functions as Python programs that seamlessly integrate with the GP framework. In what follows, we describe the four stages (initialization, reproduction, selection, termination) of our evolutionary framework. We give the formal description in Algorithm 1 – for brevity, we use $\pi$ instead of $\pi(R)$.

### 3.1 INITIALIZATION

We start by creating a population of $K$ reward function individuals using GPT-4. This leverages GPT-4's world knowledge (Hao et al., 2023) and its zero-shot instruction following abilities (Peng et al., 2023). For each individual, we train a policy $\pi$, compute a fitness score $\sigma$, and obtain natural

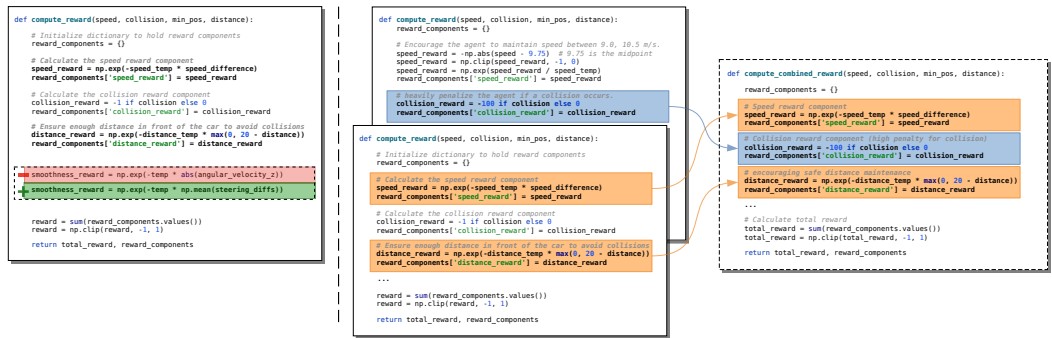

Figure 2: Illustration of how GPT-4 applies mutation and crossover to reward functions. Mutation (left): shows the modification of the "smoothness reward" component. A red '-' sign indicates the line removed from the parent reward function, while a green '+' sign indicates the line added to the new reward function. Crossover (right): demonstrates how parent reward functions are combined to create a child reward function, incorporating the most effective reward components from each parent.

language feedback $\lambda \in \mathcal{L}$. The fitness score calculation and feedback collection are detailed in Section 3.3. Each individual and their respective policy, fitness score, and natural language feedback $(R, \pi, \sigma, \lambda)$ is then randomly assigned to a sub-population and stored in the reward database $D$.

## 3.2 REPRODUCTION

Each successive generation of $K$ individuals is created by applying genetic operators (crossover and mutation), on the existing reward database individuals. We use GPT-4 as mutation and crossover operators with appropriate prompts. Due to space constraints, we provide prompt examples for mutation and crossover in Appendix E.

We begin by randomly selecting an operation – mutation with probability $p_m$ or crossover with probability $p_c = 1 - p_m$. Next, we sample a sub-population $P$ via weighted selection based on the average fitness scores $\sigma^P$ of each of the $I$ sub-populations. Subsequently, we sample parents (one for mutation, two for crossover) from $D[P]$ using weighted sampling based on individual fitness scores. These sampled individuals, along with their natural language feedback $\lambda$, are used to prompt GPT-4 to create new individuals. These prompts direct GPT-4 to either modify a single reward component of the parent ($\mathtt{Mutate}(G, D[P])$) or combine best performing reward components from two parents ($\mathtt{Crossover}(G, D[P])$). Periodically, we enable the migration of individuals between sub-populations to enhance genetic diversity. Reproduction operations are illustrated in Figure 2.

## 3.3 SELECTION

Newly reproduced individuals (from the reproduction stage) are retained based on fitness scores, following a *survival of the fittest* approach. A fitness function $F$, tailored to the specific problem, evaluates the quality of each individual. The process involves (1) training a policy $\pi(R)$ until convergence, (2) sampling rollouts $\theta \sim \Theta_{\pi(R)}$, and (3) using the fitness function to obtain a tuple of (real-valued) fitness score and (string) NL feedback $(\sigma, \lambda) := F(\theta)$. Individuals are retained if they contribute to an increase in the average fitness of their respective sub-populations $\sigma \geq \sigma^P$, where $P$ denotes the sub-population to which the individual is added[1]. This ensures that the average fitness of the sub-population is monotonically increasing. In what follows, we detail the process of obtaining the tuple $(\sigma, \lambda)$ from the fitness function.

To compute the fitness score $\sigma$, we ask human evaluators to judge policy rollouts $\theta$ from different policies in a pairwise fashion. As illustrated in the user feedback interface in Figure 1, these rollouts are presented as video segments $\approx$ 30-40 seconds long. The evaluators choose their preferred segment (ties are also possible). We then apply an Elo rating system (Elo, 1978), which transforms pairwise assessments into fitness scores $\sigma$. This avoids the need to learn a separate reward model,

---

[1]To ensure genetic diversity, we add individuals based on surpassing the average and not maximum scores.

cf. RLHF (Christiano et al., 2017). Asking human evaluators to provide preferences instead of quantitative scores for each video has the advantage of mitigating individual bias (Christiano et al., 2017). Details of the human feedback collection and Elo scoring are documented in Appendix B.1.

The NL feedback $\lambda$ about each pair of video segments is, likewise, collected through the same feedback interface. Evaluators comment (select checkboxes) on aspects of the driving they find satisfactory or need improvements. This feedback is subsequently integrated into the prompt for GPT-4 when reward function individuals are sampled for reproduction.

### 3.4 TERMINATION

The evolutionary process repeats until it reaches a predetermined number of generations $N$, or when an individual attains a fitness score equivalent to human-level driving performance, as detailed in Section 5.3. The final output is the fittest policy $\pi^*$ corresponding to the highest $\sigma$ in the database $D$. Note that $\pi^*$ is the best policy in the reward database according to human-assigned fitness scores $\sigma$, but it is not necessarily the optimal policy.

A minimal input-output example of the working of REvolve is shown in Appendix F.

## 4 RELATED WORK

**Reward Design with LLMs.** Recent works have explored LLMs for complex search and black-box optimization problems like hyperparameter optimization (Yang et al., 2024; Zhang et al., 2023; Liu et al., 2024), reward design for RL (Kwon et al., 2023; Xie et al., 2024; Ma et al., 2024a), or robotic control (Yu et al., 2023). Kwon et al. (2023) use LLMs to directly output binary rewards by inferring user intentions in negotiation games, but this requires numerous queries to LLMs and limits the rewards to sparse binary values. Conversely, Language to Rewards (L2R) (Yu et al., 2023) and Text2Reward (T2R) (Xie et al., 2024) utilize LLMs to create dense and interpretable reward functions, improving them with human feedback. L2R, however, restricts its adaptability by employing manually crafted reward templates tailored to specific robot morphologies. In contrast, T2R outputs versatile reward functions in Python code. It generates a *single* reward function per iteration, checks for improvement from the previous one, and repeats this process, hence performing a greedy search.

**Eureka** (Ma et al., 2024a) further extends T2R by generating *multiple* reward functions and *greedily* iterating with the best reward function and discarding the rest. This means that by definition (De Jong, 2017), Eureka does not qualify as an EA since it lacks crucial features. Namely, maintaining and evolving a population of candidates and employing genetic operators like mutation, crossover, and selection. Eureka only mutates the best candidate and discards the rest. In contrast, REvolve is truly evolutionary, i.e., population-based and using the full range of genetic operators. By preserving genetic diversity, REvolve avoids the pitfalls of premature convergence without incurring additional costs (since both frameworks generate the same number of candidates per generation). Empirically, we show that utilizing the full range of evolutionary operators yields reward functions that lead to better performing agents. Consequently, REvolve significantly outperforms greedy search methods like Eureka and T2R.

**Evolutionary Algorithms with LLMs.** Implementing EAs with LLMs has been explored in recent works in the context of neural architecture search (Nasir et al., 2024), program search (Romera-Paredes et al., 2024), prompt engineering (Guo et al., 2024; Fernando et al., 2023), as well as morphology design (Lehman et al., 2024). In this work, we propose a novel evolutionary framework for reward design with LLMs as intelligent genetic operators. Compared to the above-mentioned works, which feature limited evolutionary settings, REvolve incorporates a wide range of operations like mutation, crossover, selection, and migration.

## 5 EXPERIMENTS

### 5.1 EXPERIMENTAL SETUP

We performed experiments across three different environments: autonomous driving, humanoid locomotion, and adroit hand manipulation. For autonomous driving, we used the high-fidelity

Table 1: Summary of observation and action spaces, along with objectives and termination criteria for the different RL training tasks.

| Task | Observation Space | Action Space | Objective & Termination |
|------|------------------|--------------|------------------------|
| Autonomous Driving | $\mathbb{R}^{307206}$ | discrete, $|A| = 66$ | Navigate for 1000 steps |
| Humanoid Locomotion | $\mathbb{R}^{376}$ | continuous, $A \in \mathbb{R}^{17}$ | Run upright for 1000 steps |
| Adroit Hand Manipulation | $\mathbb{R}^{39}$ | continuous, $A \in \mathbb{R}^{30}$ | Open the door in 400 steps |

AirSim simulator (Shah et al., 2018). For humanoid locomotion (Tassa et al., 2012) and adroit hand manipulation (Rajeswaran et al., 2017), we used the MuJoCo simulator (Todorov et al., 2012). As a reward designer $G$, we used GPT-4 Turbo (1106-preview) model, which was provided (a subset of) observation and action variables originating from the simulators[2], as illustrated in Figure 1. The reward designer is prompted to provide a reward taking the functional form $r = \sum_i r_i$. We call the $r_i$'s reward components and the reward designer decides on their individual functional form, as well as the number of reward components. This follows the protocol established by Ma et al. (2024a).

## 5.2 Training Setup

For the evolutionary search, we set the number of generations to $N = 7$ and individuals per generation to $K = 16$. This mimics the training setup of (Ma et al., 2024a). The number of sub-populations was set to $I = 13$, and the mutation probability was set to $p_m = 0.5$.

For each generation, 16 policies were trained (one per reward function). In the AirSim environment, we trained Clipped Double Deep Q-Learning (Fujimoto et al., 2018) agents for $5 \times 10^5$ training steps per generation per agent. For the MuJoCo environments, we trained Soft Actor-Critic (Haarnoja et al., 2018) agents with $5 \times 10^6$ steps per generation per agent. We summarize the different RL tasks in Table 1 and provide detailed descriptions of the environments and RL setups in Appendix A.

Parallel training of a single generation on 16 NVIDIA A100 GPUs (40GB) consumed approximately 50 hours for the AirSim environment and 24 hours for the MuJoCo environments. Due to these extensive computational demands, we report results using two random seeds.

In addition to REvolve (with human feedback), we also include a version in the experimental evaluation that we dub **REvolve Auto**. Here, the human feedback is replaced with automatically generated feedback comprising statistics on the reward components tracked during RL training. The fitness score for each agent is now computed using a manually designed fitness function, which measures an agent's successful completion of the task. For instance, the fitness score for the humanoid locomotion task encodes that the agent should not fall over and that it should attain a high average velocity. We formally describe the fitness scores for the three different environments in Appendix B.2. This automated feedback mechanism is adopted from (Ma et al., 2024a).

## 5.3 Baselines

**Expert Designed.** For autonomous driving, we used an expert reward function based on best practices established in the literature (Spryn et al., 2018). The total reward is calculated through a weighted aggregation of all the components, with the weights optimized via an exhaustive grid search conducted within the environments to find the reward function that produces the best policy. For humanoid locomotion and adroit hand manipulation, we used expert rewards provided by Gymnasium and Gymnasium-Robotics[3], respectively. Details on the expert-designed reward functions can be found in Appendix C.

**Eureka.** We also compare REvolve to Eureka, cf. Section 4. Akin to REvolve, we employed the human fitness function to refine the best individual with natural language feedback. Henceforth, we denote it as **Eureka**. We also include the original version of Eureka with automated feedback, which we denote by **Eureka Auto**. Training times are identical to the ones for REvolve.

---

[2]To prevent potential plagiarism through context memory (Carlini et al., 2021), we use unique environment variable names and deliberately avoid mentioning the simulator in inputs to GPT-4.

[3]https://gymnasium.farama.org,https://robotics.farama.org

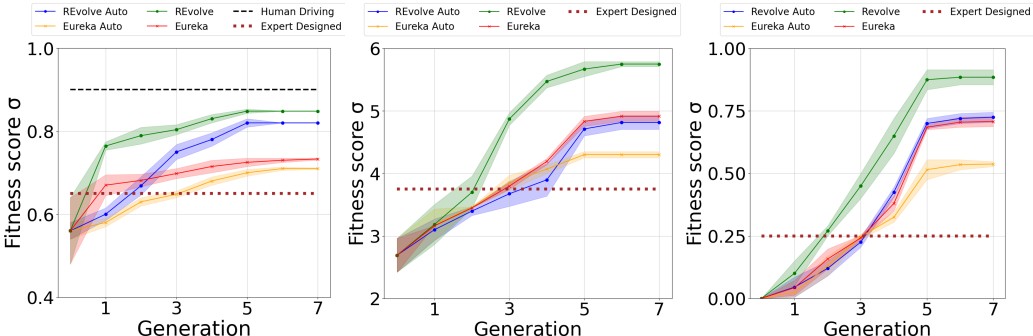

Figure 3: We plot the fitness score – over 2 random seeds – of the best-performing policy per generation for autonomous driving (left), humanoid locomotion (middle), and adroit hand manipulation (right), cf. Table 1. Each plot also contains the fitness score obtained using an expert-designed reward function. The fitness score for the human driving in the autonomous driving setting (left) was obtained by observing the driving behavior of an expert human conducting driving sessions in a combined setting of AirSim and the high-fidelity Logitech driving simulator. We treat this as the upper limit of performance. We give the formal definitions of the fitness functions for each of the three environments in Equations 2, 3, and 4, respectively (in Appendix B.2).

## 5.4 Experimental Questions & Results

As shown in Figure 3, REvolve exhibits continuous improvement over successive generations, ultimately achieving a higher fitness score on the manually designed fitness scale. Furthermore, both REvolve and REvolve Auto outperform their Eureka counterparts, indicating that evolutionary search strategies yield better results than greedy search methods[4]. We also show that all methods outperform the Expert Designed reward function. Furthermore, we observe the benefit of providing human feedback, as both REvolve and Eureka with provided human feedback outperform their automatic feedback counterparts.

**How does REvolve fair against the baselines?**
We compare REvolve to the baselines using the task-specific fitness score described in Appendix B.2 and report the results in Figure 3. We observe that REvolve (with human feedback) consistently outperforms the other methods across all three environments and is only beaten by the expert human driver in Autonomous Driving, cf. Figure 3 (left). For the interested reader, we give the best (i.e., fittest) REvolve-designed reward functions $R^*$ in Appendix G.

We also note that in the Humanoid Locomotion and Adroit Hand tasks REvolve Auto (i.e., with automated feedback) performs on par with Eureka (with human feedback), despite the former's evolutionary setting. This could be attributed to the limited and sparse nature of automated feedback (velocity for humanoid, steps to succeed for adroit hand) in contrast to the more detailed human feedback in Eureka. In contrast, automated feedback in Autonomous Driving is dense and incorporates multiple factors like collision, speed, and distance (Appendix B.2) – leading to bigger improvements during the evolutionary search. This also highlights that the combination of evolutionary search and human feedback is crucial for optimizing reward functions. It can be observed that the improvements gradually diminish with each generation and stabilize, converging by generation 5.

**How do humans judge REvolve policies?**
We further compared REvolve and Eureka on human evaluations. To this end, we generated roll-outs from the best performing policies ($\pi^*$) from REvolve and Eureka. We paired them against each other and used the user feedback interface (Figure 1) for collecting preference data. This data was subsequently passed through an Elo rating system to rank each behavior, cf. Appendix B.1. As shown in Table 2, REvolve policies again outperform Eureka policies across all three tasks, and it is only the human expert driver beating REvolve in Autonomous Driving. Note, we used a fresh batch of

---

[4]Note that these results are not displayed on the human fitness scale because the Elo ranking system evaluates individuals relative to one another. Therefore, comparisons between individuals from different baselines are only feasible if we directly pair them against one another.

Table 2: The table shows the best policies $\pi^*$ from each framework ranked based on human preferences for each environment. Specifically, we use rollouts from the best policies of each framework and gather human preferences on the paired rollouts, as described in Section 3.3. Then, we use the Elo rating system to map preferences to scores. All policies are initially assigned an Elo score of 1500.

| Environment | Rank | Baseline | Elo Score |
|---|---|---|---|
| Autonomous Driving | 1 | Human Driving | 1586 |
| | 2 | REvolve | **1575** |
| | 3 | REvolve Auto | 1529 |
| | 4 | Eureka | 1499 |
| | 5 | Eureka Auto | 1411 |
| | 6 | Expert Designed | 1397 |
| Humanoid Locomotion | 1 | REvolve | **1586** |
| | 2 | Eureka | 1557 |
| | 3 | REvolve Auto | 1514 |
| | 4 | Eureka Auto | 1434 |
| | 5 | Expert Designed | 1407 |
| Adroit Hand Manipulation | 1 | REvolve | **1594** |
| | 2 | REvolve Auto | 1524 |
| | 3 | Eureka | 1522 |
| | 4 | Eureka Auto | 1443 |
| | 5 | Expert Designed | 1415 |

evaluators (disjoint from training) for this stage. **We have submitted example rollout videos as part of the supplementary material.**

## 5.5 ABLATION STUDIES

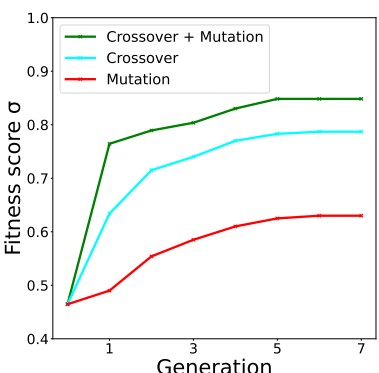

Figure 4: Comparison of REvolve's performance on Autonomous Driving using mutation only, crossover only, and both combined, over 7 generations.

Table 3: Table shows the generalization scores in unseen AirSim environments. Fitness denotes manually designed fitness scores (mean ± std). Env 1 & Env 2 have 1500 & 1000 episodic steps, respectively.

| Env | Baseline | Fitness | Episodic Steps |
|---|---|---|---|
| 1 | Expert Designed | 0.71 ± 0.03 | 710 ± 199 |
| | Human Driving | 0.95 ± 0.02 | 1500 ± 0 |
| | REvolve | **0.86 ± 0.03** | **1146 ± 297** |
| | Eureka | 0.73 ± 0.04 | 759 ± 167 |
| 2 | Expert Designed | 0.48 ± 0.02 | 71 ± 22 |
| | Human Driving | 0.75 ± 0.04 | 1000 ± 0 |
| | REvolve | **0.68 ± 0.04** | **135 ± 31** |
| | Eureka | 0.57 ± 0.05 | 99 ± 28 |

**How do different genetic operators impact overall performance?**
We evaluated REvolve's performance in Autonomous Driving by running evolution search with three setups: using only mutation, only crossover, and the combination of both. As shown in Figure 4, REvolve with mutation & crossover outperforms REvolve with only crossover, which in turn outperforms REvolve with just mutation.

**Do REvolve-designed reward functions generalize to new environments?**
Aside from our standard Autonomous Driving environment (seen in Figure 5), we assessed the generalizability of the best REvolve-designed reward functions $R^*$ in two new (AirSim) environments. Specifically, we trained policies from scratch using $R^*$ in two new scenarios: (Env 1) featuring

lanes and a completely altered landscape while maintaining similar traffic conditions as our standard environment, and (Env 2) characterized by increased traffic with multiple cars actively maneuvering. From Table 3, we observe that in both environments, REvolve outperforms Eureka and expert-designed rewards. However, the generalizability was lower for Env 2. This issue may stem from the temporal processing limitations of the RL policy's visual module, which sometimes fails to detect oncoming cars. Implementing more advanced models could enhance detection capabilities Hazra et al. (2023). Figure 6 provides visual representations of these environments, while Figure 13 provides episodic steps on the Y-axis over total steps on the X-axis during training for both generalization environments. We did not conduct similar experiments on the MuJoCo simulator, as the Humanoid and Adroit Hand environments are limited to only a single variant.

## 5.6  LIMITATIONS

As stated in Section 5.2, due to its (deep) RL nature, training multiple policies is rather computationally expensive and time-consuming. That said, the computational cost of REvolve is the same as for Eureka – with no resources wasted on discarding individuals – REvolve does incur a slightly higher storage cost due to the maintenance of the reward database (see Appendix D). Another limitation of REvolve is its reliance on the closed source GPT-4 model. This was a necessary choice as existing works have shown that a considerable gap exists in reward design abilities between GPT-4 and open-source LLMs (Xie et al., 2024; Ma et al., 2024a). With rapid advancements made by the open-source community, we believe this gap will be bridged in the near future.

## 6  CONCLUSION & FUTURE WORK

We formulated the RDP as a search problem using evolutionary algorithms for searching through the space of reward functions (Section 2). With REvolve, we proposed a novel evolutionary framework that leverages human preferences as fitness scores to guide the search process and integrates LLMs as intelligent operators to refine outputs through natural language feedback (Section 3). Our results show that this evolutionary search-based framework considerably outperforms the state of the art across three different simulation environments.

A key future question for REvolve is its real-world applicability. One promising approach involves integrating REvolve with Sim2Real algorithms Allamaa et al. (2022); Hu et al. (2024) by first learning a policy in simulation, using REvolve-designed rewards, and then transferring it to the real world. Recently, Ma et al. (2024b) have demonstrated that such an approach is viable for Eureka by tackling a quadruped balancing task on a physical robot. Moreover, our framework opens up opportunities for future research in Human-AI interaction. In particular, exploring the impact of more open-ended feedback from various perspectives, such as egocentric or third-person views.

## ETHICS STATEMENT

During human feedback experiments, we engaged 10 human evaluators per generation, each assessing 20 video pairs. Participants were recruited on a volunteer basis and the preference learning task was clearly explained to them. No personal data was collected, and participants were compensated fairly per institutional guidelines. Our study adhered to ethical standards, followed institutional guidelines, and did not involve any potential risks related to privacy, security, or discrimination.

## REPRODUCIBILITY

In the Appendix, we provide extensive details on our experimental protocol. Specific details can be found in the respective sub-appendices as detailed below. The reinforcement learning framework, including the algorithm, inputs, outputs, and hyperparameters, is outlined in Appendix A. The fitness functions are thoroughly discussed in Appendix B. The expert-designed rewards are explained in Appendix C. Detailed prompt examples for GPT-4 can be found in Appendix E. Finally, we list the best reward functions generated from REvolve in Appendix G. Our code is open-sourced at https://github.com/RishiHazra/Revolve/tree/main.

## ACKNOWLEDGMENTS

The computations and data handling for this work were enabled by the Berzelius resource at the National Supercomputer Centre, provided by the Knut and Alice Wallenberg Foundation, under projects "Berzelius-2024-178" and "Berzelius-2023- 348". This research was supported by the Wallenberg AI, Autonomous Systems, and Software Program (WASP), also funded by the Knut and Alice Wallenberg Foundation. Additionally, this work is part of the NEST Project, Multi-dimensional Alignment and Integration of Physical and Virtual Worlds (main). Finally, we thank Luc De Raedt for his valuable feedback in refining the manuscript.

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

# APPENDIX

The appendix is organized as follows: Section A Reinforcement Learning Preliminaries, where we elaborate on the reinforcement learning framework for all environments; Section B Fitness Function, where we outline the fitness score computation from human feedback, as well as outline Eureka's automatic feedback approach; Section C Expert Designed Reward Function, which provides details of the expert-designed reward and grid search parameters; Section D Cost Analysis, where we compare the computation and data requirement cost of REvolve with Eureka and RLHF on autonomous driving, as well as elaborate on the differences between REvolve and other existing approaches for autonomous driving; Section E Prompts, which presents system and operator (mutation, crossover) prompts for GPT-4 for REvolve and REvolve Auto; Section F Human Feedback Examples, which illustrates how human feedback allows GPT-4 to refine the reward function of individuals; Section G Additional Results, which lists the best reward functions generated from REvolve, and some additional results on RL convergence times and generalizability of the best reward functions.

We also include our code and demonstrations of policies trained on best REvolve-designed reward functions in the supplementary material.

## A    REINFORCEMENT LEARNING PRELIMINARIES

### A.1    ENVIRONMENTS

#### A.1.1    AUTONOMOUS DRIVING

Autonomous Driving experiments were conducted in the Neighborhood and Coastline environments of the AirSim simulator, a high-fidelity simulator developed for autonomous vehicle research (Shah et al., 2018). This urban simulation includes static features such as parked cars and dynamic obstacles like cars and moving animals, enabling a thorough evaluation of the RL agent in varied scenarios.

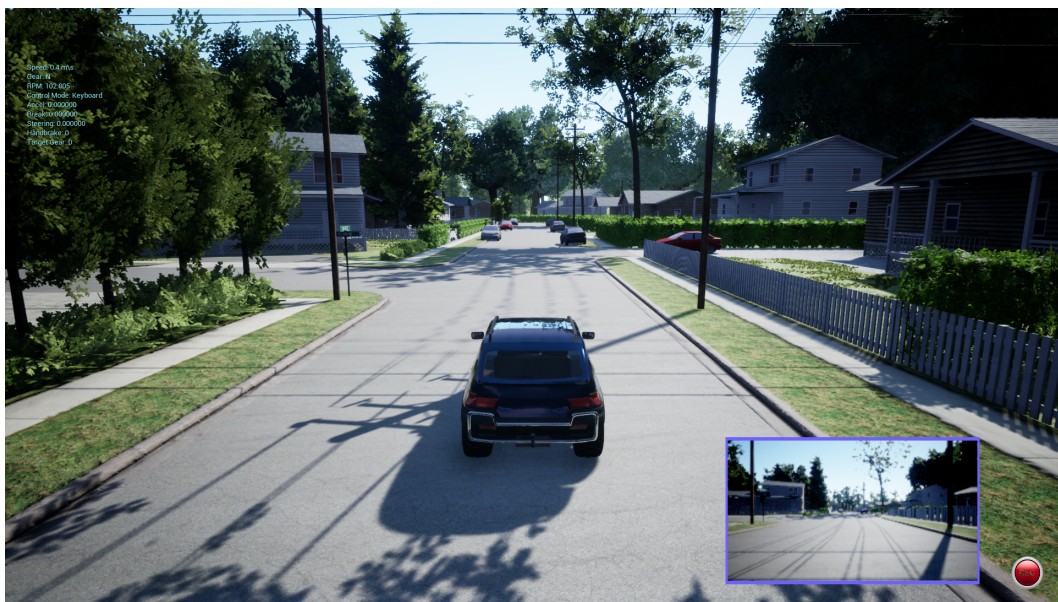

Figure 5: AirSim simulator environment used for our autonomous driving experiments. The egocentric camera view displayed on the bottom left serves as the observations for the autonomous driving agent.

**Observations.** The observation space for the policy includes a combination of visual and sensor data. The visual input consists of images concatenated along the channel axis, resulting in dimensions of $100 \times 256 \times 12$. The sensor data includes readings from yaw, pitch, linear velocity in the x and y directions, speed, and steering angle. This structure allows the network to perceive temporal dynamics by integrating sequential frames into a single input tensor.

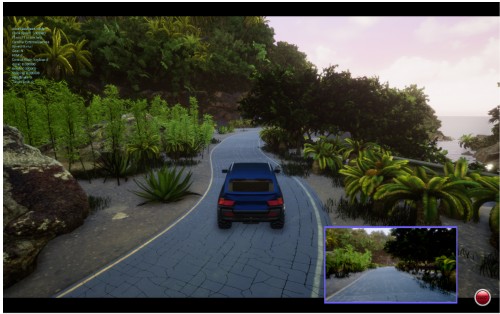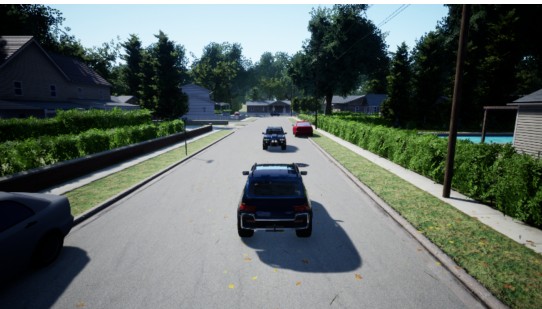

Figure 6: AirSim generalization environments used for assessing the generalizability of REvolve-designed reward functions. The left and right pictures visualize Env 1 and Env 2, respectively.

Table 4: DDQN Hyperparameters used during training for all AirSim environments for the Autonomous Driving task.

| PARAMETER | VALUE |
|---|---|
| LEARNING RATE | 0.00025 |
| OPTIMIZER | ADAM |
| GAMMA | 0.99 |
| EPSILON INITIAL | 1 |
| EPSILON MIN | 0.01 |
| EPSILON DECAY | $1 \times 10^{-3}$ |
| BATCH SIZE | 32 |
| TAU | 0.0075 |
| REPLAY BUFFER SIZE | 50000 |
| ALPHA (PER) | 0.9 |
| FREQUENCY STEPS UPDATE | 5 |
| BETA | 0.4 |
| ACTION SPACE STEERING (SIMULATOR) | [-0.8, 0.8] |
| ACTION SPACE THROTTLE (SIMULATOR) | [0,1] |
| ACTION SPACE (REAL STEERING) | [-32°, 32°] |
| IMAGE RES. | $100 \times 256 \times 12$ |
| CONV LAYERS | 5 (16-256 FILTERS) |
| DENSE UNITS AFTER FLATTEN | $256-> [128 \times 2]-> 66$ |
| SENSOR INPUT DIMENSIONS | 6 (YAW, PITCH, $V_x$, $V_y$, SPEED, STEERING) |

The input shape is $(100, 256, 12)$, and it outputs 66 discrete Q-values. To capture temporal dynamics, frame stacking aggregates four consecutive frames $[fr_1, fr_2, fr_3, fr_4]$ along the channel dimension into a single state. For subsequent states, the oldest frame is replaced by a new one, resulting in an updated stack $[fr_2, fr_3, fr_4, fr_{\text{new}}]$.

**Action Space.** The action space for the autonomous car is a 2D vector – the first dimension represents the discrete steering angles in the range $[-32°, 32°]$ in 2° increments, the second dimension is a throttling option $\{0, 1\}$. This results in 66 possible actions.

**Objective.** The agent's task is to navigate an urban environment for 1000 consecutive time steps, maintaining safe driving behavior by balancing speed and avoiding collisions.

**Termination.** The episode successfully terminates when the agent achieves the objective. Alternatively, collisions or exceeding speed limits will also prematurely end the episode.

**Training Algorithm.** Due to the high dimensional observation space and convergence time, we adopted the Clipped Double Deep Q-Learning approach (Fujimoto et al., 2018), a refined version of Double Q-Learning (Van Hasselt et al., 2016). This method uses two separate neural networks to reduce overestimation bias by calculating the minimum of the two estimated Q-values, thus providing more reliable and stable training outcomes.

In addition to Clipped Double Deep Q-Learning, our framework integrates the Dueling Architecture (Wang et al., 2016), which improves the estimation of state values by decomposing the Q-function into two streams: one that estimates the state value and another that calculates the advantages of each action. Such structure allows the model to learn which states are valuable without the need to learn each action effect for each state.

Furthermore, we utilize Prioritized Experience Replay (PER) (Schaul et al., 2015) to focus on significant learning experiences. PER prioritizes transitions with higher Temporal Difference errors for replay, which accelerates learning and increases the efficiency of the memory used during training. Table 4 lists the RL model hyperparameters. Finally, inspired by foundational research, we have adapted our architecture to incorporate vision-based methods and the Dueling Architecture from Mnih et al. (2013) and Wang et al. (2016), enhancing temporal dynamics and action-value precision.

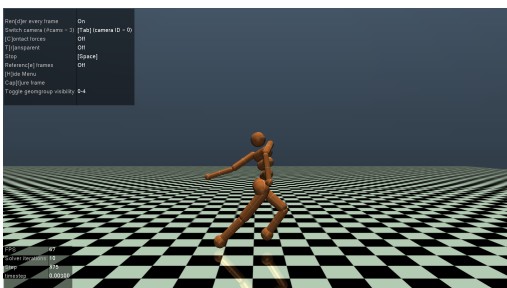

Figure 7: Humanoid Locomotion: A bipedal locomotion task where a simulated humanoid robot must maintain balance and walk as fast as possible (along the x-axis).

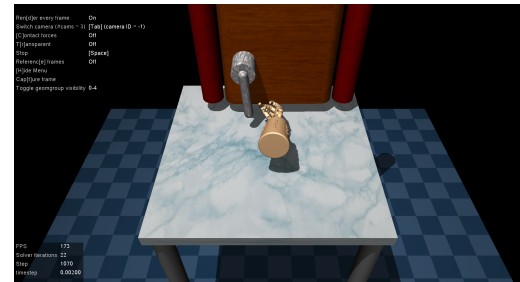

Figure 8: Adroit Hand: A complex robotic manipulation task requiring the dexterous control of a five-fingered robotic hand to rotate the latch and pull the handle to open a door.

### A.1.2 HUMANOID LOCOMOTION

The Humanoid Locomotion environment is a complex bipedal locomotion task within the MuJoCo simulator (Todorov et al., 2012).

**Observations.** The observation space for the humanoid environment includes joint positions, joint velocities, and other information such as the center of mass and contact forces. The data is crucial for controlling balance and forward locomotion. The observation shape is a 376D vector, representing the state of the humanoid at each timestep (Tassa et al., 2012).

**Action Space.** The action space is a 17D continuous vector representing the torques applied to the humanoid's joints, including those in the abdomen, hips, knees, shoulders, and elbows. This allows finer control of walking or running movements.

**Objective.** The goal is to walk or run forward without falling.

**Termination.** The episode ends either when the humanoid falls or 1000 consecutive time steps have passed without falling (success).

**Training Algorithm.** We employed the Soft Actor-Critic (SAC) algorithm (Haarnoja et al., 2018), utilizing the implementation provided by the Stable Baselines3 (SB3) library (Raffin et al., 2021). SAC is an off-policy actor-critic method that optimizes a stochastic policy in an entropy-regularized reinforcement learning framework. By maximizing a trade-off between expected return and policy entropy, SAC encourages exploration and prevents premature convergence to suboptimal policies, making it well-suited for high-dimensional continuous action spaces characteristic of MuJoCo tasks.

We leveraged the default hyperparameters provided by SB3 for SAC, which have been empirically validated across various environments. Key hyperparameters included a learning rate of $3 \times 10^{-4}$, a batch size of 256, a discount factor ($\gamma$) of 0.99, and an entropy coefficient ($\alpha$) automatically adjusted during training. The policy network employed the `MlpPolicy` architecture from SB3, consisting of multilayer perceptrons designed to process the high-dimensional state observations in MuJoCo environments effectively.

### A.1.3 ADROIT HAND

The Adroit Hand environment is another MuJoCo simulation task, which concerns dexterous robotic manipulation (Rajeswaran et al., 2017).

**Observations.** The observation space in the adroit hand environment consists of a 97D vector. It includes joint positions for the hand, latch position, door position (hinge rotation), palm position, handle position, and the relative position between the palm and the handle. Additionally, the space contains joint velocities and forces acting on the hand. The binary door open indicator is also included to signify whether the door is fully opened or closed.

**Action Space.** The action space is a 30D continuous vector that controls the absolute angular positions of the joints in the hand and arm. This enables dexterous manipulation tasks like unlatching and swinging open a door.

**Objective.** The agent's goal is to manipulate the door latch and open the door, overcoming friction and bias torque forces.

**Termination.** The episode ends when the door is successfully opened or 400 consecutive time steps have passed without opening (fail).

**Training Algorithm.** Same as humanoid locomotion.

## B FITNESS FUNCTIONS

In this section, we provide details of (1) REvolve fitness function based on human evaluations, and (2) the automatic feedback (Auto) approach proposed in Eureka.

### B.1 REVOLVE FITNESS

As stated in Section 3.3, REvolve employs a user interface to gather human feedback on behavior $t$ sampled from a trained policy. Human evaluators review pairs of videos, submitting their preferences. A fitness function then maps this data to a real-valued fitness score and a natural language feedback i.e. $(\sigma, \lambda) := F(t)$. We start by introducing the Elo rating system that maps the preference data to a fitness score. For two reward function individuals $A$ and $B$ with fitness scores $\sigma_A$ and $\sigma_B$ respectively, the expected scores $E_A$ and $E_B$ are calculated using the logistic function:

$$E_A = \frac{1}{1 + 10^{(\sigma_B - \sigma_A)/400}} \qquad E_B = \frac{1}{1 + 10^{(\sigma_A - \sigma_B)/400}}$$

We then update the individual fitness based on the expected scores.

$$\sigma_A = \sigma_A + K \cdot (f(A, B) - E_A)$$

$f(\text{X, Y})$ be defined as:

$$f(\text{X, Y}) = \begin{cases} 1 & \text{if the X wins against Y,} \\ 0 & \text{if the X loses to Y,} \\ 0.5 & \text{if the match results in a tie.} \end{cases}$$

Here, $K = 32$ determines the magnitude of fitness change. Intuitively, if higher-ranked individuals lose to those with lower rankings, they incur a more significant penalty, while the reverse scenario leads to a greater increase for the lower-ranked individual. Conversely, when two similarly ranked individuals compete, the change in fitness scores resulting from a win or loss is smaller. All individuals start with an initial rating of 1500.

For the natural language feedback $\lambda$, the checkboxes selected by evaluators – indicating aspects of the behavior that are satisfactory or require improvement – are stitched together using a natural language (NL) template. This final string serves as input for GPT-4.

The process greatly simplifies the feedback interface, as participants only need to choose their preferred video from a pair rather than providing an absolute rating. Additionally, this method

reduces individual biases that could emerge from various factors, such as the need to adjust ratings at different stages of evolution – from the initial phase, where many policies underperform, to later stages, when performances improve. To overcome this, we compare individuals not only within the same generation but also across different generations, updating the fitness scores of all individuals at the end of each generation[5]. In direct rating-based systems, participants must repeatedly rate all generations, adjusting their scales each time. In contrast, our preference-based rating system allows us to efficiently utilize all existing preference data across generations to assign new fitness scores, thus requiring minimal data collection for the current generation.

## B.2 Automatic Feedback

**Autonomous Driving.** The fitness function is primarily designed to evaluate the performance of an autonomous vehicle by assessing its behavior in speed regulation, path following, and collision occurrences. It assesses how well the vehicle adheres to speed regulations and how accurately it follows its designated waypoints while penalizing collisions and any deviations from critical operational thresholds. The function is structured as follows:

$$\sigma = \begin{cases} \text{collision penalty,} & \text{if collision occurs} \\ \text{speed penalty,} & \text{if } v < v_{\text{min\_limit}} \text{ or } v > v_{\text{max\_limit}} \\ \text{distance penalty,} & \text{if } d > d_{\text{fail}} \\ \text{score}(v, d), & \text{otherwise,} \end{cases} \tag{2}$$

where $v_{\text{adj\_min}} = v_{\text{min}} - v_{\text{threshold}}$, $v_{\text{adj\_max}} = v_{\text{max}} + v_{\text{th}}$, with $v_{\text{th}}$ small threshold speed allowing for small deviations from the target speed range.

Here, penalties are assigned for: collision penalty$= -1$ for collisions; speed penalty is calculated based on the deviation from the adjusted range $[v_{\text{adjusted\_min}}, v_{\text{adjusted\_max}}]$, decreasing as the speed moves further away from this range, with the speed score set to 0 if the speed is less than $v_{\text{min\_limit}}$ or greater than $v_{\text{max\_limit}}$; and distance penalty decreases as the vehicle moves further from the intended waypoint, with the distance score set to 0 if the deviation is above $d_{\text{fail}}$. Otherwise, the agent receives a $\text{score}(v, d) = \frac{\text{speed score}(v) + \text{distance score}(d)}{2}$.

The speed score is calculated as:

$$\text{speed score}(v) = \begin{cases} 1, & \text{if } v_{\text{adj\_min}} \leq v \leq v_{\text{max}} \\ \max(0, 1 - \frac{\min(|v - v_{\text{adj\_min}}|, |v - v_{\text{adj\_max}}|)}{v_{\text{adj\_max}} - v_{\text{adj\_min}}}), & \text{otherwise} \end{cases}$$

The speed range is $[8.0, 11.5]$ m/s. The distance score is calculated as:

$$\text{distance score}(d) = \begin{cases} 1, & \text{if } d \leq d_{\text{max}} \\ \max(0, 1 - \frac{d - d_{\text{max}}}{d_{\text{max}}}), & \text{otherwise} \end{cases}$$

with $d_{max} = 0.5$m being the maximum tolerance for deviation from the waypoint. Both functions are shown in Figure 9. In Table 5, we summarize the fitness function components used for evaluating RL models.

**Humanoid Locomotion.** The fitness function for the Humanoid environment evaluates the agent based on its ability to maintain forward locomotion without falling over an episode of $T_{\text{max}} = 1000$ steps.

$$\sigma = \begin{cases} \frac{1}{T_{\text{max}}} \sum_{t=1}^{T_{\text{max}}} v_t^x, & \text{if } T = T_{\text{max}} \\ 0, & \text{if } T < T_{\text{max}} \end{cases} \tag{3}$$

---

[5]This approach is akin to comparing chess grandmasters Gary Kasparov and Magnus Carlsen, each a dominant player in his respective era. However, a direct comparison would require them to be matched in their primes; otherwise, the basis of comparison would be skewed and unjustified.

Table 5: Fitness Function Hyperparameters used for calculating the fitness score for AirSim environment for Autonomous Driving task.

| Hyperparameter | Description | Value |
|---|---|---|
| collision_penalty | Collision penalty | -1 |
| $v_{\min}$ | Minimum speed target | 9.0 m/s |
| $v_{\max}$ | Maximum speed target | 10.5 m/s |
| $v_{\min\_limit}$ | Minimum penalty limit | 2.5 m/s |
| $v_{\max\_limit}$ | Maximum penalty limit | 15 m/s |
| $v_{adjusted\_min}$ | Adjusted minimum speed | 8 m/s |
| $v_{adjusted\_max}$ | Adjusted maximum speed | 11.5 m/s |
| $v_{th}$ | Threshold Speed | 1 m/s |
| $d_{\max}$ | Waypoint deviation tolerance | 0.5 m |
| $d_{fail}$ | Maximum tolerance for deviation from the waypoint | 4 m |

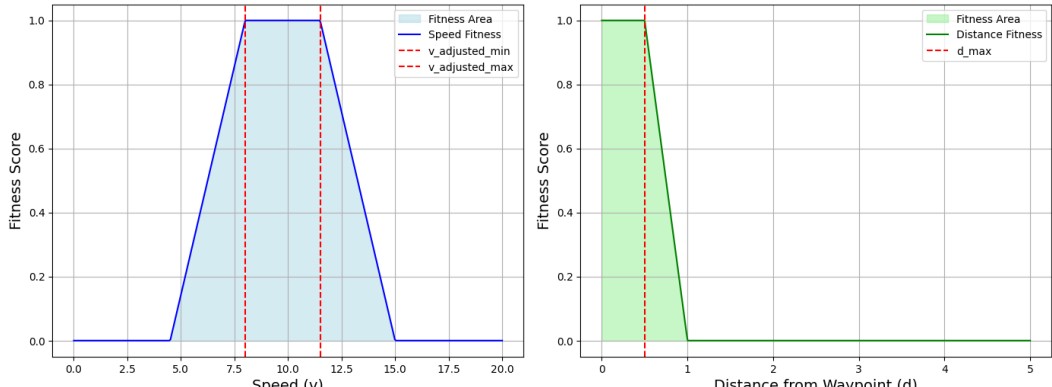

Figure 9: Visual representation of speed and distance fitness scores for autonomous driving in the AirSim environment.

where, $v_t^x$ is the forward velocity along the x-axis at time step $t$. $T$ is the total number of steps the agent remained upright. This function rewards agents that achieve higher forward velocities while also surviving the entire episode.

**Adroit Hand.** The fitness function for the Adroit Hand environment rewards agents based on the number of steps taken to successful task completion. As shown in Figure 10, completing the task in fewer steps results in a higher fitness score $\in [0.5, 1.0]$. If the task is not completed, the fitness score is 0.

Let $steps$ be the number of steps taken to complete the task ($50 \leq steps \leq 400$), $T_{\min} = 50$ be the minimum possible steps, $T_{\max} = 400$ be the maximum allowed steps, $a = -\dfrac{1}{700}, b = \dfrac{75}{70}$ are constants defining the linear relationship. The fitness score $\sigma$ is defined as:

$$\sigma = \begin{cases} a \times steps + b, & \text{if the task is successfully completed} \\ 0, & \text{if the task is not completed} \end{cases} \tag{4}$$

This score measures the efficiency of the Adroit Hand in completing the task, awarding a perfect score of 1 for completing it in 50 steps or fewer, with a linear decrease to 0.5 for completing the task in the maximum allowable 400 steps, and a score of 0 if the task is not completed.

**Feedback.** Following Eureka, we track the statistics of all reward components at intermediate policy checkpoints throughout training. We refer the interested reader to the original paper (Ma et al., 2024a).

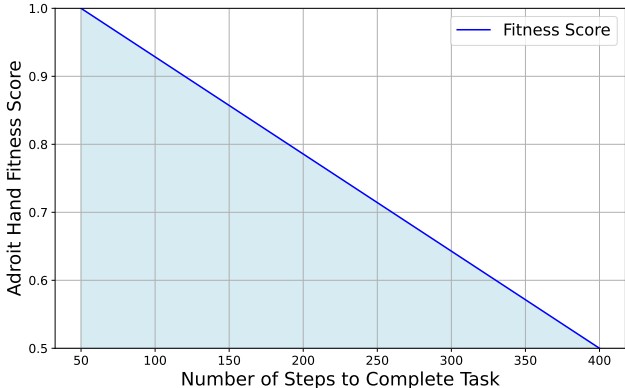

Figure 10: Visual representation of fitness score for Adroit Hand when completing the task.

Table 6: Grid Search Weights: Values and Corresponding Fitness Scores. The quadruple of weights yielding the highest fitness score is highlighted in bold.

| C_POS | C_SM | C_SPEED | C_SENSOR | FITNESS SCORE |
|---|---|---|---|---|
| 0.85 | 0.46 | 0.5 | 0.53 | 0.496 |
| 0.25 | 0.25 | 0.25 | 0.25 | 0.603 |
| 0.81 | 0.83 | 0.33 | 0.13 | 0.606 |
| 0.35 | 0.67 | 0.7 | 0.67 | 0.576 |
| 0.57 | 0.26 | 0.57 | 0.27 | 0.396 |
| 0.22 | 0.16 | 0.24 | 0.85 | 0.556 |
| 0.67 | 0.3 | 0.78 | 0.77 | 0.549 |
| 0.45 | 0.11 | 0.27 | 0.24 | 0.256 |
| 0.72 | 0.93 | 0.74 | 0.65 | 0.415 |
| 0.63 | 0.4 | 0.39 | 0.78 | 0.284 |
| **0.44** | **0.76** | **0.98** | **0.48** | **0.653** |
| 0.28 | 0.64 | 0.2 | 0.92 | 0.602 |
| 0.12 | 0.52 | 0.91 | 0.41 | 0.592 |
| 0.94 | 0.86 | 0.85 | 0.2 | 0.571 |
| 0.51 | 1.0 | 0.64 | 0.34 | 0.630 |
| 0.98 | 0.56 | 0.12 | 0.95 | 0.600 |
| 0.88 | 0.38 | 0.46 | 0.59 | 0.591 |

## C EXPERT DESIGNED REWARDS

### C.1 AUTONOMOUS DRIVING

Following the methodologies advocated in Spryn et al. (2018), the total reward at each time step, $R_{\text{total}}$, is constructed from a weighted sum of speed consistency, smooth navigation, and collision avoidance behavior of the agent. In what follows, we describe how the weights were calculated.

**Human Expert Designed (HED):** Utilizes a combination of position, speed, sensor, and smoothness rewards.

$$r_{\text{total}}^{\text{HED}} = c_{\text{pos}} \times r_{\text{pos}} + c_{\text{sm}} \times r_{\text{sm}} + c_{\text{speed}} \times r_{\text{speed}} + c_{\text{sensor}} \times r_{\text{sensor}}$$

POSITION REWARD: Encourages the vehicle to stay close to the target position, rewarding proximity to the desired path or waypoint.

$$r_{\text{pos}} = e^{-\delta \times ((\text{min\_pos})^2 - \beta)}$$

Table 7: Hyperparameters used in all reward components designed by Expert in AirSim in Autonomous Driving task.

| Hyperparameter | Description | Value |
|---|---|---|
| $\delta$ | Position reward decay factor | 0.2 |
| $\beta$ | Position reward bias | 0.25 |
| $v_{\min}$ | Minimum target speed | 9 m/s |
| $v_{\max}$ | Maximum target speed | 10.5 m/s |
| $dis_{safe}$ | Safe distance threshold | 6 m |
| $\gamma$ | Smoothness penalty coefficient | 0.5 |

SPEED REWARD: Promotes maintaining an optimal speed. It penalizes the vehicle if it moves too slowly or too quickly, encouraging it to stay within a specified speed range.

$$r_{\text{speed}} = \begin{cases} -\frac{|\text{speed} - v_{\min}|}{v_{\min}} & \text{if speed} < v_{\min} \\ -\frac{|\text{speed} - v_{\max}|}{\text{speed}} & \text{if speed} > v_{\max} \\ 1 & \text{otherwise} \end{cases}$$

SENSOR REWARD: Encourages the vehicle to maintain a safe distance from obstacles using sensor data. If the distance measured by the sensor is less than a safe threshold, it applies a penalty. The sensor measures the distance from the vehicle to nearby front objects.

$$r_{\text{sensor}} = \begin{cases} -\frac{(dis_{\text{safe}} - \text{distance})}{dis_{\text{safe}}} & \text{if distance} < dis_{\text{safe}} \\ 0.5 & \text{otherwise} \end{cases}$$

SMOOTHNESS REWARD: Penalizes abrupt changes in steering to encourage smooth driving. It uses the standard deviation of the last four steering actions to measure smoothness.

$$r_{\text{sm}} = -\gamma \times \text{std}(\text{action\_list})$$

**Human Expert Designed Tuned (HEDT):** We used grid search to optimize the values of the hyperparameters $c_{\text{pos}}, c_{\text{sm}}, c_{\text{speed}}, c_{\text{sensor}}$.

$$r_{\text{total}}^{\text{HEDT}} = c_{\text{pos}}^{opt} \times r_{\text{pos}} + c_{\text{sm}}^{opt} \times r_{\text{sm}} + c_{\text{speed}}^{opt} \times r_{\text{speed}} + c_{\text{sensor}}^{opt} \times r_{\text{sensor}}$$

These configurations allow for tailored training of autonomous driving agents, emphasizing different aspects of driving performance based on specific needs and expected driving conditions. All configurations use weights of $c_{\text{pos}}, c_{\text{sm}}, c_{\text{speed}}, c_{\text{sensor}} = 0.25$, except for the optimized version which uses customized weights.

## C.2 HUMANOID LOCOMOTION

In the humanoid environment, the agent is tasked with moving forward while maintaining a healthy state. The reward function is designed to encourage forward movement, staying upright, and minimizing control effort. The total reward $R$ is computed as:

$$R = R_{\text{forward}} + R_{\text{healthy}} - R_{\text{control}}$$

Where:

- $R_{\text{forward}} = 2.5 \times w_{fwd} \times x\_velocity$
  This term provides a reward proportional to the agent's forward velocity, incentivizing fast forward movement, with $w_{fwd} = 1.25$.

- $R_{\text{healthy}}$ is a reward for maintaining a healthy state, where the agent's height is within a predefined range:

$$R_{\text{healthy}} = \begin{cases} 15 & \text{if the agent is healthy} \\ 0 & \text{if the agent is unhealthy} \end{cases}$$

- $R_{\text{control}} = 0.5 \times w_{\text{ctrl}} \times \|\text{action}\|_2^2$
A penalty is applied based on the magnitude of the agent's actions, discouraging excessive or inefficient control efforts with $w_{\text{ctrl}} = 0.1$.

The total reward encourages the agent to move forward efficiently while staying upright and minimizing unnecessary control actions. The episode terminates if the agent becomes unhealthy (e.g., by falling) or after a maximum number of steps.

### C.3 ADROIT HAND

In the Adroit Hand Environment, the agent's task is to manipulate a door handle and open a door. The reward function is designed to incentivize both reaching the handle and successfully opening the door. The total reward $R$ is a combination of multiple components:

$$R = R_{\text{handle}} + R_{\text{door}} + R_{\text{velocity}} + R_{\text{bonus}}$$

Where:

- $R_{\text{handle}} = 0.1 \times \|\text{palm\_pos} - \text{handle\_pos}\|$
This term rewards the agent for minimizing the distance between its palm and the door handle.

- $R_{\text{door}} = -0.1 \times (\text{goal\_distance} - 1.57)^2$
This component provides a reward for moving the door hinge closer to the desired goal position (typically when the door is open).

- $R_{\text{velocity}} = -1 \times 10^{-5} \times \sum \text{qvel}^2$
A penalty is applied for excessive joint velocities (qvel), encouraging smooth and efficient movements.

- $R_{\text{bonus}}$ is a set of bonus rewards based on how much the door has been opened:

$$R_{\text{bonus}} = \begin{cases} +2 & \text{if goal\_distance} > 0.2 \\ +8 & \text{if goal\_distance} > 1.0 \\ +10 & \text{if goal\_distance} > 1.35 \text{ (goal achieved)} \end{cases}$$

The episode terminates when the door is successfully opened (i.e., goal\_distance $> 1.35$) or after 400 steps. A sparse reward version of the environment provides a reward of $+10$ for achieving the goal and $-0.1$ otherwise.

## D REVOLVE COST ANALYSIS & COMPARISONS

**Computation cost**: REvolve = Eureka. Both REvolve and Eureka output 16 reward functions and train 16 policies in each generation.

**Data requirement**: RLHF $>>$ REvolve: RLHF requires massive amounts of curated data to train a reward model. Note that the input to a reward model is a video (rollout of a policy), and the output is a preference score, requiring a heavy neural model. As a reference, the common video classification model I3D trains on the Kinetics dataset, which has 650,000 video clips (Carreira & Zisserman, 2017). IRL $>>$ REvolve: Similarly, Inverse Reinforcement Learning requires human demonstrations and interventions. In contrast, REvolve uses 10 human evaluators per generation, each assessing 20 data samples (video pairs) – hence, for 7 generations $10 \times 20 \times 7 = 1400$ curated samples only. Compare this to the Lyft self-driving dataset (Houston et al., 2021) used for motion prediction which has more than 1000 hours of data collected over 4 months by 20 cars.

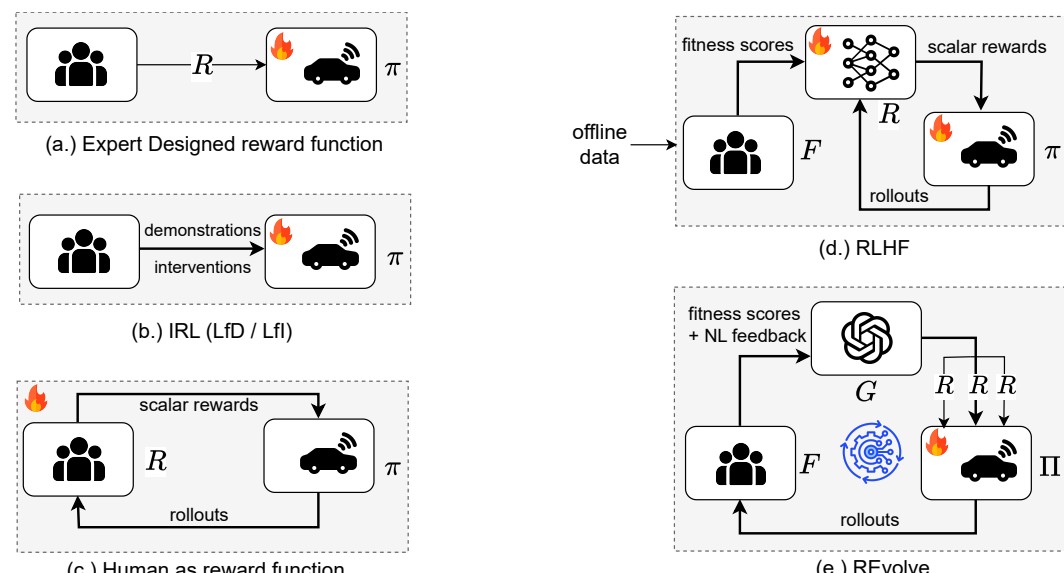

Figure 11: Comparison of REvolve with existing frameworks used to train autonomous driving agents. (a.) Expert Designed reward function $R$ to train the autonomous driving agent; (b.) learning from demonstrations (LfD) and learning from interventions (LfI), where the agent is trained to imitate the human; (c.) humans acting as reward functions within the training loop, by assessing policy rollouts and providing scalar rewards, which requires significant manual effort; (d.) Reinforcement learning from human feedback (RLHF), where human preference data is used to train an additional (black-box) reward model. This reward model is used as a proxy for human rewards to train the autonomous driving policy; (e) Proposed REvolve, which uses GPT-4 as a reward function generator $G$ and evolves them based on (minimal) human feedback on the rollouts sampled from the trained models. This feedback is directly incorporated into the reward design process. REvolve outputs interpretable reward functions, thereby avoiding learning an additional reward model. Here, $\pi \in \Pi$ is a trainable policy in the set of policies $\Pi$. The trainable blocks are denoted by the symbol of the flame.

**Storage Cost**: REvolve > Eureka. The slightly higher cost is due to maintaining a population of best candidates rather than one candidate like Eureka. However, compared to the LLM memory cost (e.g., the biggest open-source LLM Llama 3.1 takes up 810 GB of memory), the storage cost is negligible – for every generation, the storage increases by 2.5GB (which includes stored policy weights and other metadata) – 17.5 GB after 7 generations. Also, REvolve does not require expert feedback.

**Comparison with existing approaches in Reinforcement Learning and Human Feedback** Due to its complex and hard-to-specify goals (Knox et al., 2023), autonomous driving presents a challenging problem for reinforcement learning. Autonomous vehicles must maneuver through complex environments and interact with other road users, all the while strictly adhering to traffic regulations. Common techniques to overcome these challenges include learning from demonstrations (LfD) (Schaal, 1996) or learning from intervention (LfI) (Saunders et al., 2018). However, they do not always ensure that the learned behaviors align with human values such as safety, robustness, and efficiency (Hadfield-Menell et al., 2016). An alternative is integrating human feedback by using humans as the reward function in the learning process directly (Thomaz & Breazeal, 2008; Knox & Stone, 2009). However, this is rather resource-intensive as each rollout needs to be evaluated by a human. REvolve directly integrates human feedback (on the trained policy) into the reward design process. Instead of using humans as reward functions directly, human values, preferences, and knowledge can also be incorporated via so-called human-in-the-loop (HITL) RL. Several techniques exist, for instance, human assessment (Thomaz & Breazeal, 2008; Christiano et al., 2017; Knox & Stone, 2009), human demonstration (Schaal, 1996), and human intervention (Saunders et al., 2018; Wang et al., 2018). They are particularly beneficial in scenarios like autonomous driving, where the reward function is challenging, and the nature of the desired outcome is complex. More recently, reinforcement learning from human feedback (RLHF) (Christiano et al., 2017; Rafailov et al., 2023) has been used successfully in training models like ChatGPT (OpenAI, 2022), providing guidance and

alignment from user preferences. A further comparison of autonomous driving approaches with and human feedback is presented in Figure 11, which illustrates how REvolve uses human feedback to align learned policies with human driving standards without requirements for extra reward models, such as in the case of RLHF.

## E  PROMPTS

Here, we present the different prompts used for GPT-4. It includes the following:

**System Prompt** Boxes 1, 7, 10 outline the system prompts (i.e. task description and expected input from the environment, complete with formatting tips to help guide GPT-4) for generating suitable reward functions for Autonomous Driving, Humanoid Locomotion, and Adroit Hand, respectively.

**Abstracted Environment Variables** Boxes 2, 8, 11 provides a list of (abstracted) observation and action variables in the environment, formatted as a Python class, for all three environments. This informs GPT-4 of variables it can use in reward design.

**Operator Prompts** like crossover (Box 3) and mutation (Box 4). Additionally, we provide prompts for REvolve Auto baseline (Boxes 5, 6).

---

**Box 1: System Prompt: Autonomous Driving**

You are a reward engineer trying to write a reward function to create an autonomous car agent to learn to navigate the environment. Your goal is to write a reward function for the environment that will help the agent learn the task described in text.

Task description:
The objective is to create an autonomous car agent to learn to navigate the environment. Specifically, the performance metric for the learning agent is to successfully drive without crashing, while maintaining smooth driving behavior and a speed between 9.0-10.5 m/s. The episode terminates successfully if the agent drives for 1000 consecutive time steps without crashing. Otherwise, the episode terminates in failure. The controlled actions are the steering and the speed. The car must avoid staying inactive and stay as close to the center of the road as possible to ensure optimal performance.

Your reward function should use useful variables from the environment as inputs. An example reward function signature can be:
```python
def compute_reward(object_pos: type, goal_pos: type) -> Tuple[type, Dict[str, type]]:
    ...
    return reward, {}
```
The output of the reward function should consist of two items:

1. The total reward,
2. A dictionary of each individual reward component.

The code output should be formatted as a Python code string: "```python ... ```".

1. You may find it helpful to normalize to a fixed reward range like -1,1 or 0,1 by applying transformations like np.exp() to the overall reward and/or its reward components.

2. If you choose to transform a reward component, then you must introduce a temperature parameter inside the transformation function. This parameter must be a named variable in the reward function and it must not be an input variable.

3. Make sure the type of each input variable is correctly specified.

4. Most importantly, the reward's code input variables must contain only attributes of the environment. Under no circumstance can you introduce a new input variable and assume variables that are not available.

5. Your output must always be the def function code with the output being the total reward.

6. Make sure you don't give contradictory reward components.

---

**Box 2: Abstracted Environment Variables (as a Python class): Autonomous Driving**

```python
from scipy.spatial.transform import Rotation as R
import numpy as np

class Env:
    def __init__(self):
        self.client = CarClient()
        action = [CarControls().steering, CarControls().throttle]

    def state(self):
        position_info = self.client.getCarState().kinematics_estimated
        curr_x = position_info.position.x_val
        curr_y = position_info.position.y_val
        yaw, pitch = self.get_orientation(position_info)
        linear_velocity = position_info.linear_velocity
        vx, vy = linear_velocity.x_val, linear_velocity.y_val
        angular_velocity_x, angular_velocity_y, angular_velocity_z =
            position_info.kinematics_estimated.angular_velocity.x_val,
            position_info.kinematics_estimated.angular_velocity.y_val,
            position_info.kinematics_estimated.angular_velocity.z_val
        speed = position_info.speed
        collision = self.client.simGetCollisionInfo().has_collided
        action_list = [steering_value1, steering_value2,
                       steering_value3, steering_value4]
        min_pos = get_min_pos(self, position_info)
        total_step_counter = 0
        episode_step_counter = 0
        distance = 20 # default max range fo sensor is 20

    def get_orientation(self, position_info):
        quaternion = np.array([position_info.orientation.w_val,
                               position_info.orientation.x_val,
                               position_info.orientation.y_val,
                               position_info.orientation.z_val])
        euler_angles = R.from_quat(quaternion).as_euler('zyx', degrees=True)
        return euler_angles[0], euler_angles[1]

    def get_min_pos(self, position_info):
        min_dis = float('inf')
        for i in data:
            stx, sty = i[0], i[1]
            dis = eucl_dis(stx, sty, curr_x, curr_y)
            if dis < min_dis:
                min_dis = dis
        return min_dis

    def calculate_front_distance(self):
        distance_data = self.client.getDistanceSensorData(vehicle_name="Car")
        distance = distance_data.distance
        # max sensor distance is 20
```

---

**Box 3: REvolve Crossover Prompt: Autonomous Driving**

You are given a set of reward functions used to train autonomous driving agents. The reward functions have fitness scores that reflect the agent's driving performance – higher scores indicate superior driving behavior. For each reward function, we collected human feedback on what aspects of the agent's driving are satisfactory and what aspects need improvement. Additionally, we tracked the values of the individual components in the reward function after every `<episodes>` episodes and noted the maximum, mean, minimum values observed. `<EXAMPLES>`

**Your task:**

- Combine high-performing reward components from different reward functions based on human feedback and the data tracked, to enhance the agent's driving capability further.
- Clearly explain how you intend to combine them and your rationale behind improving the driving performance.
- Write the combined reward function code.

**Output of the reward function should consist of:**

- **1:** The total reward.
- **2:** A dictionary of each individual reward component.

**Code output should be formatted as a Python code string:** ```python ... ```

**Tips for reward function design:**

1. Normalize to a fixed reward range like -1,1 or 0,1 by applying transformations like np.exp() to the overall reward or its components.

2. If transforming a reward component, introduce a temperature parameter inside the transformation function. This parameter must be a named variable in the reward function and it must not be an input variable. Each transformed reward component should have its own temperature variable and you must carefully set its value based on its effect in the overall reward.

3. Ensure the type of each input variable is correctly specified.

4. The reward's code input variables must contain only attributes of the environment. Under no circumstance can you introduce a new input variable.

5. Always output the `def` function code with the output being the total reward.

6. Avoid contradictory reward components.

---

**Box 4: REvolve Mutation Prompt: Autonomous Driving**

You are given a reward function used to train autonomous driving agents. The reward functions has a fitness scores that reflects the agent's driving performance – higher scores indicate superior driving behavior. We also collected human feedback on what aspects of the agent's driving are satisfactory and what aspects need improvement. Additionally, we tracked the values of the individual components in the reward function after every `<episodes>` episodes and noted the maximum, mean, minimum values observed.

`<EXAMPLES>`

Your task is to iterate on the reward function by mutating a single component to enhance the agent's driving capability further. Some helpful tips for analyzing the reward components: If the values for a certain reward component are nearly identical throughout, then this means the training is not able to optimize this component as it is written. You may consider:

1. Rescaling it to a proper range or the value of its temperature parameter

2. Re-writing the reward component

3. Discarding the reward component

The mutation process involves the following steps:

1. First, select a reward component based on the given guidelines, human feedback, and the values of individual reward components.

2. Next, clearly explain how you intend to mutate the selected component and your rationale behind improving the driving performance. This could involve adjusting its scale, rewriting its formula, or other modifications.

3. Finally, write the mutated reward function code.

The output of the reward function should consist of two items:

1. The total reward,

2. A dictionary of each individual reward component.

The code output should be formatted as a Python code string: ` ```python ... ``` `
Some helpful tips for writing the reward function code:

1. You may find it helpful to normalize the reward to a fixed reward range like -1,1 or 0,1 by applying transformations like tf.exp() to the overall reward or its components.

2. If you choose to transform a reward component, then you must introduce a temperature parameter inside the transformation function. This parameter must be a named variable in the reward function and it must not be an input variable. Each transformed reward component should have its own temperature variable and you must carefully set its value based on its effect in the overall reward.

3. Make sure the type of each input variable is correctly specified.

4. Most importantly, the reward's code input variables must contain only attributes of the environment. Under no circumstance can you introduce a new input variable.

---

**Box 5: REvolve Auto Crossover Prompt: Autonomous Driving**

You are given a set of reward functions used to train autonomous driving agents. The reward functions have fitness scores that reflect the agent's driving performance – higher scores indicate superior driving behavior. We also tracked the values of the individual components in the reward function after every `<episodes>` episodes and noted the maximum, mean, minimum values observed. `<EXAMPLES>`

**Your task:**

- Combine high-performing reward components from different reward functions to enhance the agent's driving capability further.
- Clearly explain how you intend to combine them and your rationale behind improving the driving performance.
- Write the combined reward function code.

**Output of the reward function should consist of:**

- **1:** The total reward.
- **2:** A dictionary of each individual reward component.

**Code output should be formatted as a Python code string:** ```` ```python ... ``` ````

**Tips for reward function design:**

1. Normalize to a fixed reward range like -1,1 or 0,1 by applying transformations like np.exp() to the overall reward and/or its reward components.

2. If transforming a reward component, introduce a temperature parameter inside the transformation function.

3. Ensure the type of each input variable is correctly specified.

4. The reward's code input variables must contain only attributes of the environment.

5. Always output the `def` function code with the output being the total reward.

6. Avoid contradictory reward components.

---

---

**Box 6: REvolve Auto Mutation Prompt: Autonomous Driving**

You are given a reward function used to train autonomous driving agents. The reward function has fitness scores that reflect the agent's driving performance – higher scores indicate superior driving behavior. We also tracked the values of the individual components in the reward function after every `<episodes>` episodes and noted the maximum, mean, and minimum values observed.

`<EXAMPLES>`

Your task is to iterate on the reward function by mutating a single component to enhance the agent's driving capability further. Some helpful tips for analyzing the reward components: If the values for a certain reward component are nearly identical throughout, then this means the training is not able to optimize this component as it is written. You may consider:

1. Rescaling it to a proper range or the value of its temperature parameter

2. Re-writing the reward component

3. Discarding the reward component

The mutation process involves the following steps:

1. First, select a reward component based on the given guidelines.

2. Next, clearly explain how you intend to mutate the selected component and your rationale behind improving the driving performance. This could involve adjusting its scale, rewriting its formula, or other modifications.

3. Finally, write the mutated reward function code.

The output of the reward function should consist of two items:

1. The total reward,

2. A dictionary of each individual reward component.

The code output should be formatted as a Python code string: ```python ... ```
Some helpful tips for writing the reward function code:

1. You may find it helpful to normalize the reward to a fixed reward range like -1,1 or 0,1 by applying transformations like tf.exp() to the overall reward or its components.

2. If you choose to transform a reward component, then you must introduce a temperature parameter inside the transformation function. This parameter must be a named variable in the reward function and not an input variable. Each transformed reward component should have its own temperature variable, and you must carefully set its value based on its effect on the overall reward.

3. Make sure the type of each input variable is correctly specified.

4. Most importantly, the reward's code input variables must contain only attributes of the environment. Under no circumstance can you introduce a new input variable.

---

**Box 7: System Prompt: Humanoid Locomotion**

You are a reward engineer trying to write a reward function for humanoid robot to run. Your goal is to write a reward function for the environment that will help the agent learn the task described in text.

Task description:
The task is to train a humanoid agent to run. The robot becomes unhealthy if its torso's z-position falls outside the 1.0 to 2.0 range. The episode ends if the robot becomes unhealthy or after 1000 timesteps, with success measured by speed. Agent locomotion should be more human like.

Your reward function should use useful variables from the environment as inputs. An example reward function signature can be:
```python
def compute_reward(object_pos: type, goal_pos: type) -> Tuple[type, Dict[str, type]]:
    ...
    return reward, {}
```
The output of the reward function should consist of two items:

1. The total reward,

2. A dictionary of each individual reward component.

The code output should be formatted as a Python code string: "```python ... ```".

1. You may find it helpful to normalize to a fixed reward range like -1,1 or 0,1 by applying transformations like np.exp() to the overall reward and/or its reward components.

2. If you choose to transform a reward component, then you must introduce a temperature parameter inside the transformation function. This parameter must be a named variable in the reward function and it must not be an input variable.

3. Make sure the type of each input variable is correctly specified.

4. Most importantly, the reward's code input variables must contain only attributes of the environment. Under no circumstance can you introduce a new input variable and assume variables that are not available.

5. Your output must always be the def function code with the output being the total reward.

6. Make sure you don't give contradictory reward components.

Box 8: Abstracted Environment Variables: Humanoid Locomotion (Part 1)

```
class Env(utils.EzPickle):

    ## Action Space
    The action space is a `Box(-1, 1, (17,), float32)`. An action represents the
     torques applied at the hinge joints.

    ## Observation Space
    Observations consist of positional values of different body parts of the Humanoid,
    followed by the velocities of those individual parts (their derivatives) with all
     the
    positions ordered before all the velocities.

    By default, observations do not include the x- and y-coordinates of the torso.
     These may
    be included by passing `exclude_current_positions_from_observation=False` during
     construction.
    In that case, the observation space will be a `Box(-Inf, Inf, (378,), float64)`
     where the first two observations
    represent the x- and y-coordinates of the torso.
    Regardless of whether `exclude_current_positions_from_observation` was set to true
     or false, the x- and y-coordinates
    will be returned in `info` with keys `"x_position"` and `"y_position"`,
     respectively.

    However, by default, the observation is a `Box(-Inf, Inf, (376,), float64)`. The
     elements correspond to the following:

    | Num | Observation
                                                   | Min  | Max | Name (in corresponding XML
     file) | Joint | Unit                        |
    ---------- | ---- | --- | ------------------------------- | ----- |
     ------------------------- |
    | 0   | z-coordinate of the torso (centre)
                                                   | -Inf | Inf | root
        | free  | position (m)                |
    | 1   | x-orientation of the torso (centre)
                                                   | -Inf | Inf | root
        | free  | angle (rad)                 |
    | 2   | y-orientation of the torso (centre)
                                                   | -Inf | Inf | root
        | free  | angle (rad)                 |
    | 3   | z-orientation of the torso (centre)
                                                   | -Inf | Inf | root
        | free  | angle (rad)                 |
    ... ... ... ... ... ... ... ... ... ... ... ... ... ... ... ... ... ... ... ... ...
    | 43  | coordinate-2 (multi-axis) of the angular velocity of the angle between
     torso and left arm (in left_upper_arm)   | -Inf | Inf | left_shoulder2
          | hinge | anglular velocity (rad/s)  |
    | 44  | angular velocity of the angle between left upper arm and left_lower_arm
                                                   | -Inf | Inf | left_elbow
        | hinge | anglular velocity (rad/s)  |
    | excluded | x-coordinate of the torso (centre)
                                                   | -Inf | Inf | root
        | free  | position (m)                |
    | excluded | y-coordinate of the torso (centre)
                                                   | -Inf | Inf | root
        | free  | position (m)                |

    Additionally, after all the positional and velocity based values in the table,
    the observation contains (in order):
    - *cinert:* Mass and inertia of a single rigid body relative to the center of mass
    (this is an intermediate result of transition). It has shape 14*10 (*nbody * 10*)
    and hence adds to another 140 elements in the state space.
    - *cvel:* Center of mass based velocity. It has shape 14 * 6 (*nbody * 6*) and
     hence
    adds another 84 elements in the state space
    - *qfrc_actuator:* Constraint force generated as the actuator force. This has shape
    `(23,)`  *(nv * 1)* and hence adds another 23 elements to the state space.
    - *cfrc_ext:* This is the center of mass based external force on the body.  It has
     shape
    14 * 6 (*nbody * 6*) and hence adds to another 84 elements in the state space.
    where *nbody* stands for the number of bodies in the robot and *nv* stands for the
    number of degrees of freedom (*= dim(qvel)*)
```

---

**Abstracted Environment Variables: Humanoid Lomoton (Part 2)**

```python
    def _get_obs(self):
        device = torch.device("cuda" if torch.cuda.is_available() else "cpu")

        position = self.data.qpos.flat.copy()
        velocity = self.data.qvel.flat.copy()

        com_inertia = self.data.cinert.flat.copy()
        com_velocity = self.data.cvel.flat.copy()

        actuator_forces = self.data.qfrc_actuator.flat.copy()
        external_contact_forces = self.data.cfrc_ext.flat.copy()

        observation = np.concatenate(
        (position, velocity, com_inertia, com_velocity, actuator_forces,
 external_contact_forces,)
)

        return observation
```

---

**Box 10: System Prompt: Adroit Hand**

You are a reward engineer trying to write a reward function for the environment that will help the agent learn the task described in text.

Task description:
The task involves using an anthropomorphic robotic hand to rotate a door handle and open the door as fast as possible. The robot must manipulate the handle effectively, overcoming the friction and resistance of the door mechanism. The episode ends after 400 steps or if it opened the door successfully.

Your reward function should use useful variables from the environment as inputs. An example reward function signature can be:
```python
def compute_reward(object_pos: type, goal_pos: type) -> Tuple[type, Dict[str, type]]:
    ...
    return reward, {}
```
The output of the reward function should consist of two items:

1. The total reward,

2. A dictionary of each individual reward component.

The code output should be formatted as a Python code string: "```python ... ```".

1. You may find it helpful to normalize to a fixed reward range like -1,1 or 0,1 by applying transformations like np.exp() to the overall reward and/or its reward components.

2. If you choose to transform a reward component, then you must introduce a temperature parameter inside the transformation function. This parameter must be a named variable in the reward function and it must not be an input variable.

3. Make sure the type of each input variable is correctly specified.

4. Most importantly, the reward's code input variables must contain only attributes of the environment. Under no circumstance can you introduce a new input variable and assume variables that are not available.

5. Your output must always be the def function code with the output being the total reward.

6. Make sure you don't give contradictory reward components.

**Box 11: Abstracted Environment Variables: Adroit Hand**

```
class Env(EzPickle):
    ## Action Space

    The action space is a 'Box(-1.0, 1.0, (28,), float32)'. The control actions are
     absolute angular positions of the Adroit hand joints. The input of the control
     actions is set to a range between -1 and 1 by scaling the real actuator angle
     ranges in radians.

    ## Observation Space

    The observation space is of the type 'Box(-inf, inf, (97,), float64)'. It contains
     information about the angular position of the finger joints, the pose of the palm
     of the hand, as well as state of the latch and door and the joint velocities and
     forces.

    | Num | Observation | Min    | Max    | Joint Name (in corresponding XML file) |
     Site Name (in corresponding XML file) | Joint Type| Unit
    |-----|----------  -|--------|--------|------------|
    | 0   | Angular position of the vertical arm joint
      | -Inf   | Inf    | ARRx                                   | -
                        | hinge    | angle (rad)               |
      |
      ... ... ... ... ... ... ... ... ... ... ... ... ... ... ... ... ... ... ... ...
     ...

def _get_obs(self):
        # qpos for hand
        # xpos for obj
        # xpos for target
        qpos = self.data.qpos.ravel()
        handle_pos = self.data.site_xpos[self.handle_site_id].ravel()
        palm_pos = self.data.site_xpos[self.grasp_site_id].ravel()
        door_pos = np.array([self.data.qpos[self.door_hinge_addrs]])
        joint_velocities = self.data.qvel.ravel()
        joint_forces = self.data.actuator_force.ravel()
        if door_pos > 1.0:
            door_open = 1.0
        else:
            door_open = -1.0
        latch_pos = qpos[-1]

        return np.concatenate(
        [
            qpos[1:-2],
            [latch_pos],
            door_pos,
            palm_pos,
            palm_pos - handle_pos,
            [door_open]

    )
```

# F   HUMAN FEEDBACK EXAMPLES

---

**Box 12: Mutation with Human Feedback**

Human feedback on the autonomous driving agent's performance:
Positive: collision avoidance, lane keeping, managing consistent speed, intersection and turn handling.
Negative: Smooth Steering.

Based on the human feedback and analysis of the reward components, we can identify that the "smooth driving bonus" needs modification. The feedback indicates issues with smooth steering, which suggests that the existing smoothness component isn't effectively promoting the desired steering behavior. Given that the human feedback specifically pointed out deficiencies in this area, it suggests the measure (variance of steering actions) might not be adequately sensitive or is scaled improperly.

Intended Mutation: The mutation will focus on revising the "smooth driving bonus" to better capture and reward smooth steering. The current method uses the standard deviation of steering actions to compute a penalty, which may not effectively discriminate between slightly varying and highly erratic steering actions. A more responsive approach might involve squaring the deviation before scaling, which penalizes larger variations more severely and provides a sharper gradient for the learning algorithm to optimize against.

Rationale By squaring the deviation, small steering adjustments are less penalized, allowing for natural minor corrections while driving. Larger, more erratic movements result in significantly higher penalties, pushing the training to favor smoother transitions and more stable steering behaviors.

```python
def old_compute_reward(curr_x: float, curr_y: float, speed: float, collision: bool,
    action_list: list, total_step_counter: int, episode_step_counter: int, min_pos:
    float, distance: float) -> Tuple[float, Dict[str, float]]:
    ...

    max_steering_variance = 0.1
    smoothness_factor = np.std(action_list)
    reward_components['smooth_driving_bonus'] = np.exp(-temp_smoothness * (
    smoothness_factor / max_steering_variance))
    ...

def compute_reward(curr_x: float, curr_y: float, speed: float, collision: bool,
    action_list: list, total_step_counter: int, episode_step_counter: int, min_pos:
    float, distance: float) -> Tuple[float, Dict[str, float]]:
    ...

    # Updated mutation for smooth driving bonus
    max_steering_variance = 0.1
    smoothness_factor = np.std(action_list)
    reward_components['smooth_driving_bonus'] = np.exp(-temp_smoothness * (
    smoothness_factor**2 / max_steering_variance**2))
    ...
```

---

# G   ADDITIONAL RESULTS

## G.1   BEST PERFORMING REWARDS

### G.1.1   AUTONOMOUS DRIVING

In Box 13, we present the best-performing reward from REvolve with a fitness score of $0.84$ for the AirSim environment. Similarly, Box 14 displays the best-performing reward from REvolve Auto with a fitness score of $0.81$ for the same environment.

**Box 13: Best reward function from REvolve: Autonomous Driving**

```python
def compute_reward(curr_x: float,
                   curr_y: float,
                   speed: float,
                   collision: bool,
                   min_pos: float,
                   action_list: List[float]) -> Tuple[float, Dict[str, float]]:

    # Parameters to tweak the importance of different reward components
    collision_penalty = -100
    inactivity_penalty = -10
    speed_reward_weight = 2.0
    position_reward_weight = 1.0
    smoothness_reward_weight = 0.5

    # Adjusted temperature parameters for score transformation
    speed_temp = 0.5  # Increased temperature for speed_reward
    position_temp = 0.1
    smoothness_temp = 0.1

    reward_components = {
        "collision_penalty": 0,
        "inactivity_penalty": 0,
        "speed_reward": 0,
        "position_reward": 0,
        "smoothness_reward": 0
    }

    # Penalize for collision
    if collision:
        reward_components["collision_penalty"] = collision_penalty

    # Penalize for inactivity (speed too close to zero)
    if speed < 4.5:  # Adjusted threshold for inactivity
        reward_components["inactivity_penalty"] = inactivity_penalty

    # Reward for maintaining an optimal speed range
    if 9.0 <= speed <= 10.5:
        speed_score = 1 - np.abs(speed - 9.75) / 1.75  # Centering and normalizing
     around the average of optimal speed range
    else:
        speed_score = -1
    # Use a sigmoid function to smoothly transform the speed score and constrain it
    reward_components["speed_reward"] = speed_reward_weight * (1 / (1 + np.exp(-
     speed_score / speed_temp)))

    # Reward for being close to the center of the road
    position_score = np.exp(-min_pos / position_temp)
    reward_components["position_reward"] = position_reward_weight * position_score

    # Reward for smooth driving (small variations in consecutive steering actions)
    steering_smoothness = -np.std(action_list)
    reward_components["smoothness_reward"] = smoothness_reward_weight * np.exp(
     steering_smoothness / smoothness_temp)

    # Calculate total reward, giving precedence to penalties
    total_reward = 0
    if reward_components["collision_penalty"] < 0:
        total_reward = reward_components["collision_penalty"]
    elif reward_components["inactivity_penalty"] < 0:
        total_reward = reward_components["inactivity_penalty"]
    else:
        total_reward = sum(reward_components.values())

    # Ensure the total reward is within a reasonable range
    total_reward = np.clip(total_reward, -1, 1)

    return total_reward, reward_components
```

Box 14: Best reward function from REvolve Auto: Autonomous Driving

```python
def compute_reward(curr_x: float, curr_y: float, speed: float, collision: bool,
    action_list: list, total_step_counter: int, episode_step_counter: int, min_pos:
    float, distance: float) -> Tuple[float, Dict[str, float]]:
    # Define temperature parameters for reward transformations
    temp_collision = 5.0
    temp_inactivity = 10.0
    temp_centering = 2.0
    temp_speed = 2.0
    temp_smoothness = 1.0
    temp_proximity = 2.0

    # Initialize the total reward and reward components dictionary
    reward_components = {
        'collision_penalty': 0,
        'inactivity_penalty': 0,
        'lane_centering_bonus': 0,
        'speed_regulation_bonus': 0,
        'smooth_driving_bonus': 0,
        'front_distance_bonus': 0
    }

    # Penalty for collisions
    reward_components['collision_penalty'] = np.exp(-temp_collision) if collision else
     0

    # Penalty for inactivity
    inactivity_threshold = 4.5
    if speed < inactivity_threshold and not collision:
        reward_components['inactivity_penalty'] = np.exp(-temp_inactivity * (
     inactivity_threshold - speed))

    # Lane centering bonus
    max_distance_from_center = 2.0  # adaptable based on road width
    reward_components['lane_centering_bonus'] = np.exp(-temp_centering * min_pos)

    # Speed regulation bonus
    ideal_speed = (9.0 + 10.5) / 2
    speed_range = 10.5 - 9.0
    reward_components['speed_regulation_bonus'] = np.exp(-temp_speed * (abs(speed -
     ideal_speed) / speed_range))

    # Smooth driving bonus
    max_steering_variance = 0.1  # configured to the acceptable steering variance for
     full bonus
    smoothness_factor = np.std(action_list)
    reward_components['smooth_driving_bonus'] = np.exp(-temp_smoothness * (
     smoothness_factor / max_steering_variance))

    # Front distance bonus
    max_distance_view = 20
    reward_components['front_distance_bonus'] = np.clip(distance / max_distance_view,
     0, 1)

    # Sum the individual rewards for the total reward
    total_reward = sum(reward_components.values())

    # Ensure penalties take precedence
    if collision or speed < inactivity_threshold:
        total_reward = -1  # Apply max negative reward for collision or inactivity

    # Normalize the total reward to lie between -1 and 1
    total_reward = np.clip(total_reward, -1, 1)

    return total_reward, reward_components
```

### G.1.2 HUMANOID LOCOMOTION

In Box 15, we present the best-performing reward from REvolve with an average velocity of 5.67. In Box 16, the best-performing reward from REvolve Auto has an average velocity of 4.71.

Box 15: Best reward function from REvolve: Humanoid Locmotion

```python
def compute_reward(observation: np.array) -> (float, dict):
    # Constants and Parameters
    speed_temp = 0.1
    orientation_temp = 0.05
    health_temp = 1.0

    # Extract relevant components from the observation
    speed = observation[22] # x-velocity component for reward
    z_position = observation[0] # z-coordinate of the torso
    orientation = observation[1:5] # orientation of the torso (x, y, z, w)

    # Reward Components
    speed_reward = np.exp(speed * speed_temp) - 1  # Encourage forward movement,
     exponentially
    orientation_penalty = -np.var(orientation) * orientation_temp  # Penalize variance
     in orientation to encourage stability
    health_penalty = 0 if 1.0 < z_position < 2.0 else -np.exp(-abs(z_position - 1.5) *
     health_temp)  # Penalty for unhealthy z-position

    # Total Reward Calculation
    total_reward = speed_reward + orientation_penalty + health_penalty

    reward_components = {
        'speed_reward': speed_reward,
        'orientation_penalty': orientation_penalty,
        'health_penalty': health_penalty
    }

    return total_reward, reward_components
```

Box 16: Best reward function from REvolve Auto: Humanoid Locomotion

```python
def compute_reward(observation: np.ndarray) -> (float, dict):
    # Constants and Parameters
    speed_reward_temp = 0.1
    health_penalty_temp = 0.01
    orientation_reward_temp = 0.1
    health_threshold = 0.8  # Assumed threshold for the health component before
     penalties kick in more aggressively

    # Extracting necessary components from observation for computation
    z_position = observation[0]  # Assuming first value represents torso z-position
    speed = observation[22]  # Assuming this represents the forwards velocity

    # Speed Reward Component: Encourages forward movement
    speed_reward = speed * np.exp(speed_reward_temp * speed)

    # Health Penalty Component: Penalizes unhealthy states less aggressively
    is_unhealthy = 1 if z_position < 1.0 or z_position > 2.0 else 0
    health_penalty = -np.exp(health_penalty_temp * is_unhealthy * (2.0 - z_position) if
      z_position < health_threshold else health_penalty_temp * is_unhealthy * 0.5)

    # Orientation Reward Component: Encourages maintaining upright torso orientation
    x_orientation, y_orientation = observation[1], observation[2]  # Assumed
     orientation metrics
    orientation_reward = (1 - x_orientation**2 - y_orientation**2) * np.exp(
     orientation_reward_temp * (1 - x_orientation**2 - y_orientation**2))

    # Total Reward Calculation
    total_reward = speed_reward + health_penalty + orientation_reward

    # Reward Components Dictionary
    reward_components = {
        'speed_reward': speed_reward,
        'health_penalty': health_penalty,
        'orientation_reward': orientation_reward
    }

    return total_reward, reward_components
```

### G.1.3 ADROIT HAND

"In Box 17, we present the best-performing reward from REvolve with a fitness score of 0.81. In Box 18, the best-performing reward from REvolve Auto has a fitness score of 0.64."

---

**Box 17: Best reward function from REvolve: Adroit Hand**

```python
def compute_reward(observation, joint_velocities, joint_forces):
    # Example observables
    goal_pos = observation[28]  # Using door_pos as an approximation for the goal
     position
    palm_to_handle = np.linalg.norm(observation[35:38])  # Magnitude of vector from
     palm to handle
    door_hinge_angle = observation[28]
    completion_threshold = 1.35  # radians, as per task success definition

    # Parameters
    dist_temp = 10.0  # Temperature parameter for distance cost transformation
    hinge_temp = 5.0  # Temperature parameter for hinge reward transformation

    # Components mutation: Distance cost modification with exponential decay
    dist_cost = -np.exp(-palm_to_handle / dist_temp)

    # Door hinge reward: reward for rotating the door handle
    door_hinge_reward = np.exp((door_hinge_angle - completion_threshold) / hinge_temp)
     if door_hinge_angle < completion_threshold else 1.0

    # Checking if the door is sufficiently opened to consider the episode a success
    if door_hinge_angle >= completion_threshold:
        completion_bonus = 100.0  # Significant bonus for completing the task
    else:
        completion_bonus = 0.0

    # Total reward calculation
    total_reward = dist_cost + door_hinge_reward + completion_bonus

    # Returning the total reward and the individual components as a dictionary
    return total_reward, {'dist_cost': dist_cost, 'door_hinge_reward':
     door_hinge_reward, 'completion_bonus': completion_bonus}
~
```

---

**Box 18: Best reward function from REvolve Auto: Adroit Hand**

```python
def compute_reward(observation: np.ndarray) -> (float, dict):
    # Defining temperature parameters for the transformations
    temp_position = 0.1  # Temperature for position reward transformation
    temp_door_open = 0.05  # Temperature for door open reward transformation

    # Extract necessary components from the observation
    position_reward = observation[-2]  # Assuming last but one value corresponds to
     positional info
    door_open_reward = observation[-1]  # Assuming last value indicates door being open
      or not

    # Apply transformations with temperature parameters
    transformed_position_reward = np.exp(temp_position * position_reward) - 1  #
     Subtracting 1 to ensure a baseline of 0
    transformed_door_open_reward = np.exp(temp_door_open * door_open_reward) - 1

    # Calculate total reward
    total_reward = transformed_position_reward + transformed_door_open_reward

    # Constructing the dictionary of individual reward components
    reward_components = {
        'position_reward': transformed_position_reward,
        'door_open_reward': transformed_door_open_reward
    }

    return total_reward, reward_components
```

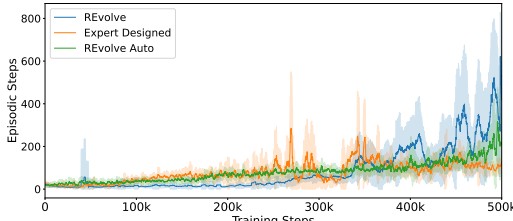 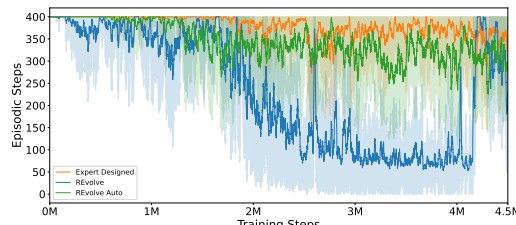

Figure 12: Compares policy training with REvolve, REvolve Auto and expert-designed reward functions on episodic steps from the start of the training for the Autonomous Driving [Left] and Adroit Hand [Right]. Note, that episodic steps in Autonomous Driving denote how long an agent drives (i.e. higher is better). In contrast, episodic steps in Adroit Hand denote the number of steps the agent takes to solve the tasks (lower is better).

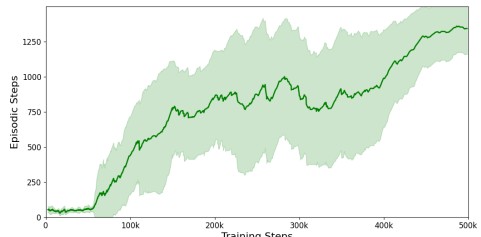 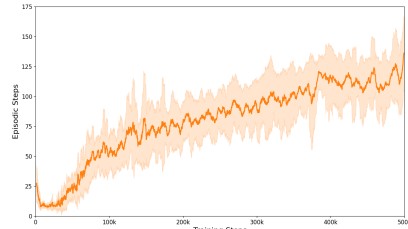

Figure 13: Episodic Steps during training for Env 1 [Left] and Env 2 scenario [Right].

**Do REvolve reward functions lead to higher success rates?** In Figure 12, we see that REvolve-designed rewards converge to a higher number of episodic steps compared to expert-designed rewards, signifying a better success rate per episode – higher is better for Autonomous Driving and vice versa.

