# OpenReview forum: "REvolve: Reward Evolution with Large Language Models using Human Feedback"
_ICLR.cc/2025/Conference — ICLR 2025 Poster_

### Official Review · Reviewer_vtL6 · 2024-10-29

**Soundness:** 3
**Presentation:** 4
**Contribution:** 3
**Rating:** 6
**Confidence:** 3

**Summary:**

This paper introduces a reward design framework. It proposes a fitness function that involves human feedback and natural language. It applies evolutionary algorithms to evolve a population of reward functions by LLMs, where the prompt is the output of the fitness function.

**Strengths:**

REvolve uses the evolutionary algorithm with genetic operators like mutation, crossover, and selection to overcome the disadvantage in greedy search in Eureka and present good performance.
REvolve leverages LLMs to generate reward function explicitly and employ human preference to guide the reward function generation implicitly, which enhances the interpretability and aligns with humans.

**Weaknesses:**

No major weaknesses under the topic of explicit reward function generation, but it lacks the discussion of the comparison to an implicit reward design, for example, a reward model from human preference.

**Questions:**

(1) In Figure 3 (left), it seems auto grows faster than human feedback. In Figure 3 (right), auto grows exponentially but others grow linearly. What will happen given more generations? Will the auto surpass human feedback? I would suggest to run more generations to see a clear trend.

(2) A follow up question for (1). If the auto surpass human feedback, or even human driving, would human evaluators still prefer human feedback to auto?

(3) To solve reward design problem, REvolve is a good solution for explicit reward function generation, but how does this method compare with implicit reward models, i.e., RLHF, in terms of performance, interpretability, computational requirements?

---

> ### Author Response · Authors · 2024-11-15
> **Response to Reviewer vtL6**
>
> We thank the reviewer for their time and effort to review and provide us with insightful feedback. We greatly appreciate for acknowledging our contribution of *fully evolutionary, human feedback for better alignment, interpretable reward functions*.
>
> We answer your questions here. **Please refer to the Common questions where we provided updated results beyond 5 generations, as requested**, which answers Questions (1), (2)
>
> ```(3) To solve reward design problem, REvolve is a good solution for explicit reward function generation, but how does this method compare with implicit reward models, i.e., RLHF, in terms of performance, interpretability, computational requirements?```
> We refer the reviewer to Appendix D, Page 24 where we contrast REvolve with RLHF and Inverse Reinforcement Learning. To summarize, (1) Data and computational requirements: RLHF requires massive amounts of curated data to train a reward model. Note that the input to a reward model is a video (rollout of a policy), and the output is a preference score, requiring a heavy neural model; (2) Interpretability: As stated in L81-86, REvolve generates interpretable reward functions whereas RLHF uses black-box reward models.
>
> More broadly, as stated in Section 2, our work addresses the *reward design* problem, which involves explicitly specifying a reward function. In contrast, RLHF focuses on *reward learning*, where the reward function is inferred from data or feedback. These are fundamentally distinct approaches.

---

> > ### Comment · Reviewer_vtL6 · 2024-11-15
> >
> > Thanks for the answers. I have some follow up questions about the updated results. It seems all settings become saturated after 5 iterations and they achieve to their own saturated value. What is the explanation of this phenomenon? Specifically, why do different methods reach to the saturation almost at the same time (after 5 iterations). Second, why does each method have their own saturation point? Is it related to certain inherent properties of evolutionary algorithms?

---

> ### Author Response · Authors · 2024-11-16
> **Author response to follow-up questions**
>
> We thank the reviewer for engaging with us. We would be happy to address any further questions or concerns raised by the reviewer.
>
> ```Specifically, why do different methods reach to the saturation almost at the same time (after 5 iterations)```
>
> We clarify that we do not interpret these figures as fully converging after 5 generations. Instead, upon examining the scores of each baseline from the autonomous driving and humanoid locomotion tasks, it is evident that for autonomous driving, REvolve and Eureka begin to converge much earlier compared to the *Auto baselines. Here, we define convergence as no change in the first decimal place.
>
> REvolve:         [0.56, 0.76, 0.79, **0.80, 0.83, 0.85, 0.85,0.85**]
> Eureka:           [0.56, 0.67, 0.68, **0.70, 0.72, 0.73, 0.73, 0.73**]
> REvolve auto: [0.56, 0.60, 0.67, 0.75, 0.78, **0.82, 0.82, 0.82**]
> Eureka auto:   [0.56, 0.58, 0.63, 0.65, 0.68, **0.70, 0.71, 0.71**]
>
> Similarly, for the humanoid locomotion task, REvolve, Eureka, and REvolve Auto begin to converge by the 6th generation, while Eureka Auto achieves convergence earlier, in the 5th generation.
>
> REvolve:         [2.69, 3.18, 3.70, 4.87, 5.47, 5.67, **5.75, 5.75**]
> Eureka:           [2.69, 3.16, 3.45, 3.80, 4.19, 4.83, **4.92, 4.92**]
> REvolve Auto: [2.69, 3.10, 3.40, 3.67, 3.90, 4.71, **4.82, 4.82**]
> Eureka Auto:   [2.69, 3.18, 3.46, 3.85, 4.07, **4.30, 4.30, 4.30**] ​
>
>
> For the Adroit Hand task, we observe an abrupt change in performance at generation 5. For REvolve, this can be attributed to the system approaching the upper limit of the fitness function (=1). However, for the other baselines, we currently lack a clear intuition to explain this abrupt change. That said, **the observed gaps in the saturation points across different baselines are not unexpected**. This is explained in our next answer.
>
> REvolve:         [0.00, 0.10, 0.27, 0.45, 0.65, **0.88, 0.88, 0.88**]
> Eureka:           [0.00, 0.04, 0.16, 0.24, 0.38, 0.68, **0.70, 0.71**]
> REvolve Auto: [0.00, 0.04, 0.12, 0.22, 0.43, **0.70, 0.72, 0.72**]
> Eureka Auto:   [0.00, 0.02, 0.13, 0.25, 0.32, **0.52, 0.54, 0.54**]
>
> ```Second, why does each method have their own saturation point? Is it related to certain inherent properties of evolutionary algorithms?```
> **Yes!**
> Comparing REvolve & Eureka saturation points: As discussed in L303-307, the difference in saturation points can be attributed to the evolutionary approach of REvolve, which outperforms the greedy approach of Eureka in complex optimization problems. The evolutionary method excels because it explores a broader range of solutions, increasing the likelihood of discovering near-global optima, whereas greedy search tends to get trapped in local optima.
>
> Comparing REvolve/Eureka & REvolve/Eureka Auto saturation points: As highlighted in L408-420, the use of a human fitness function combined with natural language feedback proves to be more effective for open-ended tasks. This leads to higher scores and, consequently, higher saturation points for both REvolve and Eureka compared to their automated counterparts.
>
> Finally, our experimental analysis demonstrates that the best performance arises from the combination of an evolutionary approach and human feedback. This is evident from the higher saturation point of REvolve compared to all baselines.

---

> > ### Author Response · Authors · 2024-11-20
> > **Author Follow up**
> >
> > Dear Reviewer,
> >
> > Once again, thank you for engaging with us. We would be happy to answer any further questions/issue clarifications if needed.

---

### Official Review · Reviewer_CReE · 2024-11-04

**Soundness:** 3
**Presentation:** 4
**Contribution:** 3
**Rating:** 6
**Confidence:** 4

**Summary:**

REvolve is an evolutionary algorithm for LLM reward function generation for RL tasks in robotics. Authors first point out that Eureka is not in fact an evolutionary algorithm (despite claiming to be so), because it does not preserve populations and do mutation and crossovers. REvolve takes Eureka and implements two main changes: make it an actual evolutionary algorithm, and use human labelers as a fitness function (instead of a manually-specified one as is done by Eureka). Authors demonstrate REvolve surpasses Eureka on 3 sim tasks: autonomous driving, humanoid locomotion, and adroit hand door opening.

**Strengths:**

1. Paper is written very clearly and is easy to understand.

2. Results included good ablations that tested the effect of human ratings as fitness functions for both REvolve and Eureka.

3. Human evaluations were conducted and full Elo scores reported between REvolve, Eureka, and their respective ablations.

4. Appendix contained detailed experimental setup information.

**Weaknesses:**

1. Natural language feedback to the LLM is extracted from a series of checkboxes that the user ticks. This requires a human to predefine the specific characteristics that the agent must exhibit, which may bias the prompting of the model by constraining the natural language feedback to this small discrete list of checkboxes. So while the authors argue that no manual engineering is required for providing the fitness function, there is manual engineering required to provide a good list of attributes for the human labelers to choose from, and it seems like these attributes must be tailored for each task.

2. A more in-depth analysis that would be helpful to the paper would try to quantify the total amount of human effort needed in REvolve vs Eureka auto: amount of time to define the natural language feedback checklist, provide all pairwise ratings and natural language feedback throughout training in REvolve, vs time to define a fitness function in Eureka. Since it requires hundreds of human labels over N=5 generations, I suspect REvolve involves much more human effort than Eureka auto, but the promising part of this work is that REvolve auto outperforms Eureka auto on all three tasks.

3. Presumably there is additional effort involved in REvolve vs Eureka, from a ML practitioner standpoint, in needing to tune additional parameters related to the mutation and crossover operations (such as p_m, or number of islands).

**Questions:**

1. What proportion of the LLM-generated reward functions were executable? Eureka mentioned many generated reward functions could not be executed.

2. Each human provides feedback consisting of a pairwise preference assessment and natural language feedback (extracted from a selection of checkboxes). This seems somewhat demanding. How important is each of these two components, relatively? Ablating either of the two components would have been informative.

3. Did authors try allowing humans to provide open-form natural language feedback, not constrained to a checklist?

4. Were the human evaluators for the policy’s final performance in Table 2 the same as the human evaluators that provided feedback during training? If so, I would take issue with that, because then the non-auto methods are given an unfair advantage over the auto methods since they were evaluated on the same distribution of human evaluators as during training.

5. During crossover, how are the “most effective” reward components determined? Through human natural language feedback?

6. Perhaps show an evaluation comparison as training proceeds through the N=5 generations. This would tell us whether REvolve consistently outperforms Eureka, or whether REvolve learns faster with increasing generation index.

7. What is the feasibility of REvolve on real robot tasks? Authors mentioned sim2real as a promising approach, but given the difficulty in simulating high DOF robots (such as humanoids), training on simulation data may not necessarily yield satisfactory results. However, only training with real data would be expensive with REvolve especially with the high demand on human feedback.

---

> ### Author Response · Authors · 2024-11-15
> **Response to Reviewer CReE**
>
> We thank the reviewer for taking and time and effort to provide us feedback. We appreciate that the reviewer finds our paper *clearly written* with *good ablations including human evaluations*, and a *detailed appendix*.
>
> We address your queries here. **Please also refer to the common question at the top for the updated results from 7 generations, as requested.**
>
> ```Manual engineering required to provide a good list of attributes for the human labelers to choose from, and it seems like these attributes must be tailored for each task.```
> We acknowledge that some manual engineering is necessary to tailor the list of attributes, particularly in our current setup. However, this is a **one-time effort that brings substantial benefits**. By predefining relevant attributes, we significantly reduce the time and cognitive load during the human feedback stage. Instead of crafting and typing detailed natural language feedback, evaluators can simply select from predefined checkboxes, streamlining the process.
>
> Furthermore, **specifying high-level attributes for improvement is a much less demanding cognitive task than designing a comprehensive fitness function**. Crafting such a function would require evaluators to account for all possible factors while assigning specific weights to each, which is both time-consuming and complex.
>
> ```Quantify the total amount of human effort needed in REvolve vs Eureka auto -- and that REvolve might involve much more human effort than Eureka auto```
> Indeed, the human effort required for REvolve is more than Eureka auto, **however that is justified by the significant improvement in performance**. As stated in Appendix D, REvolve uses 10 human evaluators per generation, each assessing 20 data samples (video pairs) – hence, for 7 generations 10 × 20 × 7 = 1400 curated samples. This is not required for auto feedback.
>
> More generally speaking, **providing automated feedback for complex open-ended tasks (like autonomous driving) might not be very practical**. As we highlight in L 54-56, and noted by ```Reviewer zYk2``` -- if such a fitness function were available, it might logically replace the reward function altogether. This necessitates an alternate approach like a human-centered one where fitness is inferred through human preference rather than predefined metrics. That said, the **preference data requirement is still significantly lower than what would be required for a RLHF-like framework** (see  Appendix D).
>
> ```Presumably there is additional effort involved in REvolve vs Eureka, from a ML practitioner standpoint, in needing to tune additional parameters related to the mutation and crossover operations (such as p_m, or number of islands).```
> In practice, while an ML practitioner cares about the complexity and resource demands of additional hyperparameters, **the ultimate value is often judged by the quality of the end result**. Typically, practitioners are willing to invest time in hyperparameter tuning if it means achieving a meaningful improvement in performance, which in our case, it does.
>
> **Notably, the REvolve hyperparameters were task-agnostic**, i.e. no task-specific tuning was required and we used the same hyperparameters across all.
>
> ```What proportion of the LLM-generated reward functions were executable? ```
> In each generation, we run 16 LLM-generated reward functions. Approximately ~15% of these functions result in errors, requiring the LLM to generate a new output. The most common errors stem from the LLM hallucinating variables that are not part of the environment.
>
> ```Feedback consisting of a pairwise preference assessment and natural language. How important is each of these two components, relatively? ```
> Both the pairwise preference assessment and natural language feedback play distinct and indispensable roles in our approach. The preference assessment provides the fitness scores that drive the evolutionary algorithm’s core steps (selection, migration, etc.). Meanwhile, the natural language feedback directs the LLM operators on which specific aspects to prioritize during mutation and crossover.
>
> Note that Eureka also incorporates both (1) automated feedback (e.g., statistics of all reward components at intermediate policy checkpoints) to inform operators, and  (2) fitness scores in selecting and mutating the top-performing individual. The statistics intuitively signal whether each reward component is improving over time, guiding the LLM on which components may need adjustment. Although we briefly mention this in L 1132-1133 of the Appendix, we point the reviewer to the original Eureka paper for details.
>
> **Without fitness scores, the evolutionary approach would fail, and without feedback, the LLM operators would operate blindly.**

---

> ### Author Response · Authors · 2024-11-15
> **Response to Reviewer CReE (Part 2)**
>
> ```...allowing humans to provide open-form natural language feedback?```
> Excellent point! We’ve considered this approach and would like to explore it further within the REvolve framework. However, it presents several additional challenges: (1) Subjectivity – Open-ended feedback varies widely due to individual interpretations shaped by experience, cultural norms, and personal risk tolerance; (2) Data Bias – Models may overfit to specific feedback patterns, especially if biases exist within the participant pool; (3) Utility – Unstructured responses risk introducing noise that could hinder meaningful behavior adjustments.
>
> Given these challenges, we believe this area merits dedicated research and consider it a valuable direction for future work, as stated in L 520-522.
>
> ```Were the human evaluators for the policy’s final performance in Table 2 the same as the human evaluators that provided feedback during training? ```
> **No**, we used different sets of evaluators for both. We've clarified this in the main paper (L 454-455). Thank you!
>
> ```During crossover, how are the “most effective” reward components determined? Through human natural language feedback?```
> **Yes!** After the selection stage, natural language feedback is used to guide the LLM operators to combine of the most effective reward components. It’s important to note that these reward components aren’t necessarily aligned with the attribute checkboxes which are merely to provide general behavioral cues. The LLM must use its commonsense knowledge to map the feedback to specific reward components.
>
> ```What is the feasibility of REvolve on real robot tasks?```
> Sim2real transfer of Eureka has already been demonstrated in Dr Eureka [1] on the Unitree Go1 robot for a quadrupedal locomotion task. Given the superior performance of REvolve and its generalizability, the exact framework, should in principle applicable for REvolve too.
>
> [1] DrEureka: Language Model Guided Sim-To-Real Transfer, Ma et al., RSS 2024

---

> > ### Author Response · Authors · 2024-11-20
> > **Follow up by Authors**
> >
> > Dear Reviewer,
> >
> > We were wondering if there are any further questions or clarifications you'd like us to address. We'd gladly provide additional details if needed.
> >
> > Thank you for your time and consideration.

---

> > ### Comment · Reviewer_CReE · 2024-11-27
> > **Response to Author's Rebuttal**
> >
> > Thank you for taking the time to answer my questions line-by-line!
> >
> > Everything makes sense, but I'm not sure if I buy the second-half of the argument that: "Without fitness scores, the evolutionary approach would fail, and without feedback, the LLM operators would operate blindly."
> >
> > For instance, we could only collect preference assessments from humans (without natural language checkboxes), and simply tell the LLM that the human preferred demonstration A over demonstration B. We could then query the LLM to take in as input videos/trajectory representations of demonstrations A and B and summarize the differences in language. Then we can provide this language difference description to REvolve as a proxy for the human checkboxes, so the approach has a signal for how to tweak the reward function. It likely won't work as well as your current approach, but I believe this kind of ablation could feasibly perform somewhat well. However, I responded too late to ask for additional experiments (sorry), so perhaps you can explore this in the future if it sounds interesting to analyze.

---

> > > ### Author Response · Authors · 2024-11-28
> > > **Thank you for the interesting suggestion**
> > >
> > > Dear Reviewer,
> > >
> > > Thank you for the suggestion. You raise an interesting point!
> > >
> > > Simplifying REvolve’s feedback mechanism by using human preference signals and querying the LLM to infer behavior differences from trajectories would also largely automate the framework (seeing the big picture here). But as you've insightfully pointed out, this approach (of using LLMs to summarize the videos) may currently face practical limitations, as recent studies [1, 2], suggest:
> > >
> > > * **Spatial-Temporal Reasoning Deficits:** As shown in [1], models like GPT-4V achieve only 49.3% accuracy on spatial-temporal tasks compared to 80.1% for humans, indicating that LLMs struggle to deduce meaningful causal or contextual differences from complex demonstrations.
> > >
> > > * **Visual Reasoning Gaps in Multimodal Models:** VLMs tend to rely more on textual inputs and struggle to effectively interpret visual information, especially in tasks requiring reasoning across trajectories [2]. This could hinder the LLM’s ability to generate actionable feedback from demonstration comparisons alone.
> > >
> > > As multimodal models mature and their ability to infer temporal and spatial patterns improves, such mechanisms could significantly simplify REvolve’s pipeline while maintaining strong performance. We will explore these ideas in future iterations.
> > >
> > > ---
> > >
> > >
> > > [1] ConTextual: Evaluating Context-Sensitive Text-Rich Visual Reasoning in Large Multimodal Models. arXiv:2401.13311
> > > [2] Is A Picture Worth A Thousand Words? Delving Into Spatial Reasoning for Vision Language Models. arXiv:2406.14852

---

### Official Review · Reviewer_zYk2 · 2024-11-04

**Soundness:** 3
**Presentation:** 4
**Contribution:** 2
**Rating:** 6
**Confidence:** 3

**Summary:**

This paper presents a framework that combines human feedback with genetic algorithms to guide LLMs in designing complex reward functions for agents. The authors conducted experiments on three tasks: Autonomous Driving, Humanoid Locomotion, and Adroit Hand Manipulation, and performed a comprehensive ablation study analysis.

**Strengths:**

- Well written, the technical approach is very clear, especially Figure 1 and Algorithm 1, which can help readers quickly understand the technical route of the paper.
- The experiments involve a wide range of task types, covering multiple dimensions of continuous and discrete action spaces and observation spaces for RL agents in virtual simulation environments.
- The authors have compared their work with a variety of baseline algorithms.

**Weaknesses:**

The paper does not clearly describe the foundational setup of the experiments and the comparison metrics. In fact, I do not understand the role of the "fitness score" mentioned in the text. In sections 2 and 3, it is used as a supervisory signal to guide the genetic algorithm in generating the reward function. However, in section 4, it becomes an evaluation metric for the experimental results of the paper. I am unsure whether this practice is appropriate because it raises the following concerns:
- (1) If the fitness score can effectively measure the performance of agents in the current task, then why not use it directly as the reward function to train the agents?
- (2) If the fitness score serves as a signal to guide the generation of the reward function, is it fair to use it as an evaluation metric for the experiments? This is because the REsolve method receives fitness supervision during training, whereas other baseline methods do not.
The insights provided by the paper are limited. When the RLHF paradigm was introduced, people saw that using human feedback could enable agents to perform challenging actions like backflips, which are certainly beyond the capabilities of rule-based reward functions. However, in the tasks and experimental setup implemented in the paper, all I understand is that the authors have used very costly human feedback to achieve a performance superior to the baselines, without offering additional insights or analyses to justify the necessity of this approach.

**Questions:**

- The first sentence of the paper mentions "Recent success...", but the cited articles are from 2018 and 2019. Is that really considered recent?
- How long does the entire pipeline process take? Are there cases where the reward function does not converge due to difficulties in summarizing it from human-provided language feedback? I think this is an important issue because the pipeline proposed in this paper seems overly complex and involves a lot of uncertainties.

---

> ### Author Response · Authors · 2024-11-15
> **Response to Reviewer zYk2**
>
> We appreciate the Reviewer's time and effort in reviewing our paper. We appreciate that the Reviewer believes our paper is *well-written* with a *clear technical approach*, and a *diverse range of experiments*.
>
> We address your potential misunderstanding about the fitness score, reward function, and evaluation metrics here.
>
> ```(1) If the fitness score can effectively measure the performance of agents in the current task, then why not use it directly as the reward function to train the agents?```
> **Indeed, this aligns with our argument in L 54-57 for preferring REvolve over Eureka.**
>
> Let's start by noting that designing an intricate fitness function, especially for open-ended tasks like autonomous driving, is itself challenging. As insightfully pointed out by the reviewer, if such a fitness function were available, it might logically replace the reward function. Hence, it is indeed impractical to assume access to such a fitness function as in Eureka. **We address this dilemma in REvolve by bypassing the need for a predesigned fitness function**, by mapping human preferences directly to Elo scores (aka fitness scores), **effectively using humans as the fitness function** (L 75-79). Our Elo rating system is detailed in Section 3.3.
>
> However, in practice, one could also use a much simpler fitness function to measure the agent's performance along certain concrete axes (Appendix B.2). Of course, this would not comprehensively cover every single detail of the agent's behavior, thus affecting the stages of the evolutionary process -- for instance, it is possible to term an agent as fit just because the fitness function measures collisions, even if the agent overspeeds or driving unsafely. We empirically demonstrate that through our REvolve auto baseline where the fitness scores of the agents are determined through a predesigned fitness function. Note, how REvolve (with human fitness) outperforms REvolve auto. This demonstrates the utility of using human feedback within an evolutionary setting. Additionally, REvolve auto helps us post a fair comparison with Eureka auto which also employs predesigned fitness functions.
>
> ```(2) If the fitness score serves as a signal to guide the generation of the reward function, is it fair to use it as an evaluation metric for the experiments?```
> Great point! **We, in fact, employ multiple evaluation metrics** beyond just the fitness function, including human assessments of trained policies (Table 2), and episodic steps — how long agents avoid collisions or overspeeding (Table 3). As stated, our predesigned fitness functions are simple, tracking collisions, speed penalties, and lane alignment, for instance.
>
> **That said, in RL, it's not uncommon to evaluate policies on how high their rewards are at convergence -- the same rewards that is used to train them**. This is especially relevant for open-ended tasks like autonomous driving, which lack clear-cut objectives, unlike structured games such as Chess or Go.
>
> ```"Recent success...", but the cited articles are from 2018 and 2019.```
> We have added more recent citations too (L32-33). Thank you.
>
> ```How long does the entire pipeline process take?```
> The breakdown is as follows: For each generation (1) LLM outputs 16 individuals: ~10 minutes; (2) RL training which is dependent on the task. It can range from ~24 hours for Adroit to ~48 hours for autonomous driving (L349-351); (3) Human feedback which takes ~2 hours. Notably, REvolve and Eureka have the same run-time and use the same computational resources (see Appendix D).
>
> ```Are there cases where the reward function does not converge due to difficulties in summarizing it from human-provided language feedback?```
> **No**. As shown by the standard deviation in Figure 3, **REvolve seeds converge toward similar values, particularly in the final generations.**

---

> > ### Comment · Reviewer_zYk2 · 2024-11-17
> > **Response by Reviewer zYk2**
> >
> > I am very grateful for the author's timely and detailed response, which accordingly improved my score.

---

### Author Response · Authors · 2024-11-15
**Updated Figures over 7 generations.**

We sincerely thank all the reviewers for their valuable feedback, which has greatly helped us improve our work. We are elated that the reviewers acknowledge our core contributions (```vtL6,CReE```) and find our paper **well-written** (```zYk2,CReE```), our **technical approach clear** (```zYk2```), our **experiments diverse** (```zYk2,CReE```), and our **appendix detailed** (```CReE```).

In response to questions raised by reviewers ```CReE``` and ```vtL6``` regarding experiments beyond five generations, we have revised the paper and updated Figures 3 and 4 to reflect results for **seven generations instead of five**. The findings are as follows:

* All baselines **converge/stabilize around generation 5** across all tasks (Figure 3, L421-423) .
* At convergence, REvolve continues to significantly outperform all the other baselines including Eureka and human-engineered rewards. This is also qualitatively evident from our supplementary videos which were part of the original submission.
* Human feedback methods (REvolve and Eureka) show clear advantages over automated feedback approaches (REvolve Auto and Eureka Auto), leading to better overall performance.

---

### Meta-Review · Area_Chair_dbXe · 2024-12-19

**Metareview:**

This paper proposes a novel algorithm for tuning RL policies to optimize human preferences. Their framework uses an LLM to generate reward functions from natural language descriptions, and then uses genetic algorithms to mutate these functions to iteratively improve performance. Compared to the closest prior work, Eureka, there are two main contributions: (i) the "true" reward requires soliciting human feedback rather than running a simulator, and (ii) using genetic algorithms instead of a simpler greedy iterative refinement approach. While the novelty is somewhat limited, the contribution still appears sufficiently interesting to merit acceptance.

**Additional Comments On Reviewer Discussion:**

The reviewers had some concerns about clarity and about the experimental setup, which were partially addressed during the rebuttal period. Overall, they agree that the paper is sufficiently interesting to merit acceptance.

---

### Decision · Program_Chairs · 2025-01-22

Accept (Poster)